# Revealing the distribution of transmembrane currents along the dendritic tree of a neuron from extracellular recordings

**Dorottya Cserpán[1], Domokos Meszéna[2,3], Lucia Wittner[2,4], Kinga Tóth[2], István Ulbert[2,3,4], Zoltán Somogyvári[1,4,5], Daniel K Wójcik[6]\***

[1]Wigner Research Centre for Physics, Hungarian Academy of Sciences, Budapest, Hungary; [2]Institute of Cognitive Neuroscience and Psychology, Research Centre for Natural Sciences, Hungarian Academy of Sciences, Budapest, Hungary; [3]Faculty of Information Technology and Bionics, Pázmány Péter Catholic University, Budapest, Hungary; [4]National Institute of Clinical Neurosciences, Budapest, Hungary; [5]Neuromicrosystems Ltd., Budapest, Hungary; [6]Department of Neurophysiology, Nencki Institute of Experimental Biology of Polish Academy of Sciences, Warsaw, Poland

**\*For correspondence:**
d.wojcik@nencki.gov.pl

**Competing interests:** The authors declare that no competing interests exist.

**Abstract** Revealing the current source distribution along the neuronal membrane is a key step on the way to understanding neural computations; however, the experimental and theoretical tools to achieve sufficient spatiotemporal resolution for the estimation remain to be established. Here, we address this problem using extracellularly recorded potentials with arbitrarily distributed electrodes for a neuron of known morphology. We use simulations of models with varying complexity to validate the proposed method and to give recommendations for experimental applications. The method is applied to in vitro data from rat hippocampus.
DOI: https://doi.org/10.7554/eLife.29384.001

## Introduction

A variety of methods are used to investigate the electrophysiological properties of neurons. To date, patch-clamp (*Neher and Sakmann, 1976*) is the most commonly used technique to monitor neuronal membrane potential. Despite its unquestionable utility, it remains challenging to monitor the activity of a cell at more than one or two sites. Extracellular recordings, on the other hand, deliver a more global picture of neural activity (*Buzsáki et al., 2012*; *Einevoll et al., 2013a*). With modern multielectrodes and microelectrode arrays, it is now possible to record neuronal activity from many thousands of channels (*Buzsáki, 2004*; *Berdondini et al., 2005*; *Obien et al., 2014*). However, this technique does not permit direct recording of membrane potentials but instead spiking activity [which may be of individual cells (single-unit activity, SUA) or multiple cells (multiunit activity or MUA, which is the mean firing rate of cell populations)] and components of postsynaptic activity visible at low frequencies (<300 Hz, so-called local field potential, LFP); see (*Buzsáki et al., 2012*; *Einevoll et al., 2013a*; *Głąbska et al., 2017*) for discussion.

So far, the main advantages of high-density array recordings have been improved resolution of spike detection (*Rey et al., 2015*), as more cells can be identified in a single recording, improved stimulation precision (*Hottowy et al., 2012*; *Chichilnisky, 2001*), of particular importance for retinal neuroprosthetics, and new features observed in the profiles of slow fields (*Ferrea et al., 2012*).

Recently, high-density probes have also been used in studies of axon tracking (*Bakkum et al., 2013*; *Lewandowska et al., 2016*) and multisynaptic integration (*Jäckel et al., 2017*).

Multielectrode recordings have been traditionally used for improved spike sorting (*Buzsáki, 2004*; *Berdondini et al., 2005*; *Obien et al., 2014*; *Muthmann et al., 2015*) and for the reconstruction of current source densities (CSD) behind the recorded LFP (*Pitts, 1952*; *Mitzdorf, 1985*; *Wójcik, 2015*), although more specific methods have sinse been devised (*Einevoll et al., 2013b*; *Głąbska et al., 2014*; *Głąbska et al., 2016*). Several attempts have been made, using different approaches, to localize cells from multielectrode recordings. For example, accounting for the properties of electric field propagation in the tissue (*Muthmann et al., 2015*), that form the basis of CSD methods (*Somogyvári et al., 2005*, Somogyvári et al., 2012*Somogyvári et al., 2012*), or other triangulation approaches (*Mechler et al., 2011*; *Mechler and Victor, 2012*). We are not aware of any prior attempts, however, to reconstruct the CSD of individual neurons using their available morphologies, which we propose here.

This method assumes we have a set of extracellular recordings, coming from a specific neuron, whose morphology and location with respect to the electrode is known, collected with multiple contacts. This could be realized experimentally by patching a neuron close to the multielectrode and driving it through an intracellular injection or monitoring its activity to determine the contribution of this specific cell to the extracellular field. Computing spike-triggered average of the potential, which we do in our proof-of-concept experiment, or driving the neuron with sinusoidal current and averaging the extracellular potential over periods of the driving current, are ways in which this could be achieved. When the recordings are complete, we inject dye into the cell and reconstruct its morphology. Thus, we obtain a set of synchronous multichannel extracellular recordings reflecting the activity of a single neuron whose morphology is also known, as well as the position of the neuron relative to the electrode contacts. Here, we show how to use this approach to infer the distribution of current sources based on cell morphology as they change in time. The data necessary to apply presented method have been available for some time (*Henze et al., 2000*; *Gold et al., 2006*), although recently have became much more comprehensive (*Jäckel et al., 2017*). While we believe the estimation of transmembrane currents along the cell morphology using this type of data has not been reported previously, similar questions have been posed by (*Gold et al., 2006*), who attempted to identify the biophysical properties of a neuron membrane based on the extracellular signature of the action potential. A similar strategy was used by *Frey et al., 2009*) in their studies of extracellular action potential shape observed with high-definition multi-electrode arrays.

The *single-cell kernel Current Source Density* method (skCSD) we introduce here is an application of the framework of the *kernel Current Source Density method* (*Potworowski et al., 2012*) to the data coming from a single cell. This is done by restricting current sources to cell morphology. This can be done efficiently for arbitrarily complex morphologies and arbitrary electrode configurations.

The importance of this work is that for the first time we show here how a collection of extracellular recordings in combination with cell morphology can be used to estimate how the current sources located on a studied cell contribute to the recorded field potential. Since it is feasible to acquire the relevant data, we believe that the method proposed here may be used to constrain the biophysical properties of the neuron membrane and facilitate consistency of the reconstructed morphology. Further, this method can help guide new discoveries by providing a more global picture of the current distribution based on neuronal morphology, leading to a coherent spatiotemporal view of synaptic drive and return currents of the observed neuron.

In the following Results section, we start with a high-level overview of the skCSD method. We explain how it is applied and why it works. Then, we validate this method on several ground truth datasets obtained in simulations and apply it to data from a proof-of-concept experiment. In the Discussion, we summarize our main findings and discuss the practical aspects and feasibility of experimental acquisition of the required data. Finally, in the Materials and methods section, we present the skCSD method in detail.

## Results

The main result of this work is the introduction of the skCSD method, so we start here with a high-level overview. The technical details are deferred to the Materials and methods section. Next, we

study the properties of the skCSD reconstruction for three representative morphologies of increasing complexity and for different setups.

First, for a ball-and-stick neuron, we study the general quality of reconstruction of fine detail by considering CSD distributions in the form of standing waves of increasing spatial frequency which form the Fourier basis of any possible CSD profile. It is unlikely that standing waves would be naturally observed in a cell, therefore to better understand how the results for the Fourier space representation relate to a specific distribution which might arise in a physiological situation, we also consider reconstruction of sources for random synaptic activation of the ball-and-stick cell.

Secondly, we consider a Y-shaped neuron with a single branching point, we check if skCSD can differentiate between synaptic activations located on the different branches close to the branching point. We also investigate the effects of random distribution of contacts on skCSD reconstruction. Finally, we investigate the possibility of skCSD reconstruction on a realistic model of a ganglion cell placed on a microelectrode array (MEA) as well as the sensitivity of the method to noise.

After establishing and validating skCSD on these fully controlled model datasets, to demonstrate neurophysiological viability, the CSD distribution was reconstructed for a pyramidal cell using the experimental spike-triggered averages of the recorded potentials.

## Single-cell kernel current source density method, a high-level overview

The goal of this section is to provide an overview of the proposed method for non-specialists, with limited mathematical terminology and notation. A more formal discussion is provided in the Materials and methods section.

Assume we study a neuron, we have its morphology, we know how it is located with respect to a multielectrode used for extracellular recording, we also have a set of simultaneous recordings of extracellular potential generated by this cell collected with this multielectrode. In principle, the number and placement of the electrodes can be arbitrary. Also, the potential may be filtered, or we may consider the full spectrum of the signal, depending on whether we wish to focus more on synaptic contributions or consider the extracellular signatures of spiking. For now, we shall ignore the challenge of separating the part of the signal contributed by the studied neuron from background extracellular signals generated by nearby cells; we shall return to this issue in the Discussion.

We wish to reconstruct the distribution of current sources which generated the measured potentials. By assumption, we know the potential that comes from the studied cell, we wish to restrict the sources to lie on its morphology. To do this, we first represent the morphology by a closed line which we call the *morphology loop*. To construct it consider a one-dimensional abstraction of the cell, where we ignore the thickness of the dendrites. Alternatively, you may imagine the graph constructed from the lines passing centrally through all the dendrites. Then, starting for example at the soma, we draw a line along this graph passing all sections along the dendrite, eventually reaching the starting point. The morphology loop is shown as the red line in *Figure 1*.

If we spread the morphology loop, we obtain a circle, which means all point of dendritic morphology have been mapped to a circle, and the opposite, any point on the circle has been mapped to the morphology. The mapping from the cell to the circle is not unique: we pass by most points on the dendrite twice, with the exception of the tips, which are visited once, and the branching points, which may be visited more than twice in our graph representation. So in most cases, a given point from the morphology corresponds to multiple points from the circle. The other mapping is unique: every point in the circle is mapped onto exactly one point on the morphology without skipping any. One way to think about the morphology loop is as a rubber band tightly wrapping around the neuron's morphology.

We now want to consider the distribution of current sources on the morphology. We found it convenient, technically, to start with a distribution of sources along the morphology loop. Then, we wrap this distribution around the cell together with the loop. We do this by construction. We cover the morphology loop with a large number of identical but translated functions which we call the *CSD basis functions* denoted by $\widetilde{b}_j(\mathbf{x})$. There is a large flexibility here, but in practice we use Gaussians, so $\widetilde{b}_j(\mathbf{x}) \propto \exp(-(\mathbf{x} - \mathbf{x}_j)^2/2R^2)$. The number of basis functions we use, $M$, and the *width of the basis function*, $R$, are parameters of the method whose effect on results we discuss below.

We place these basis functions so that they uniformly cover the morphology loop. We require their centers to be uniformly spaced. When we wrap these functions around the morphology,

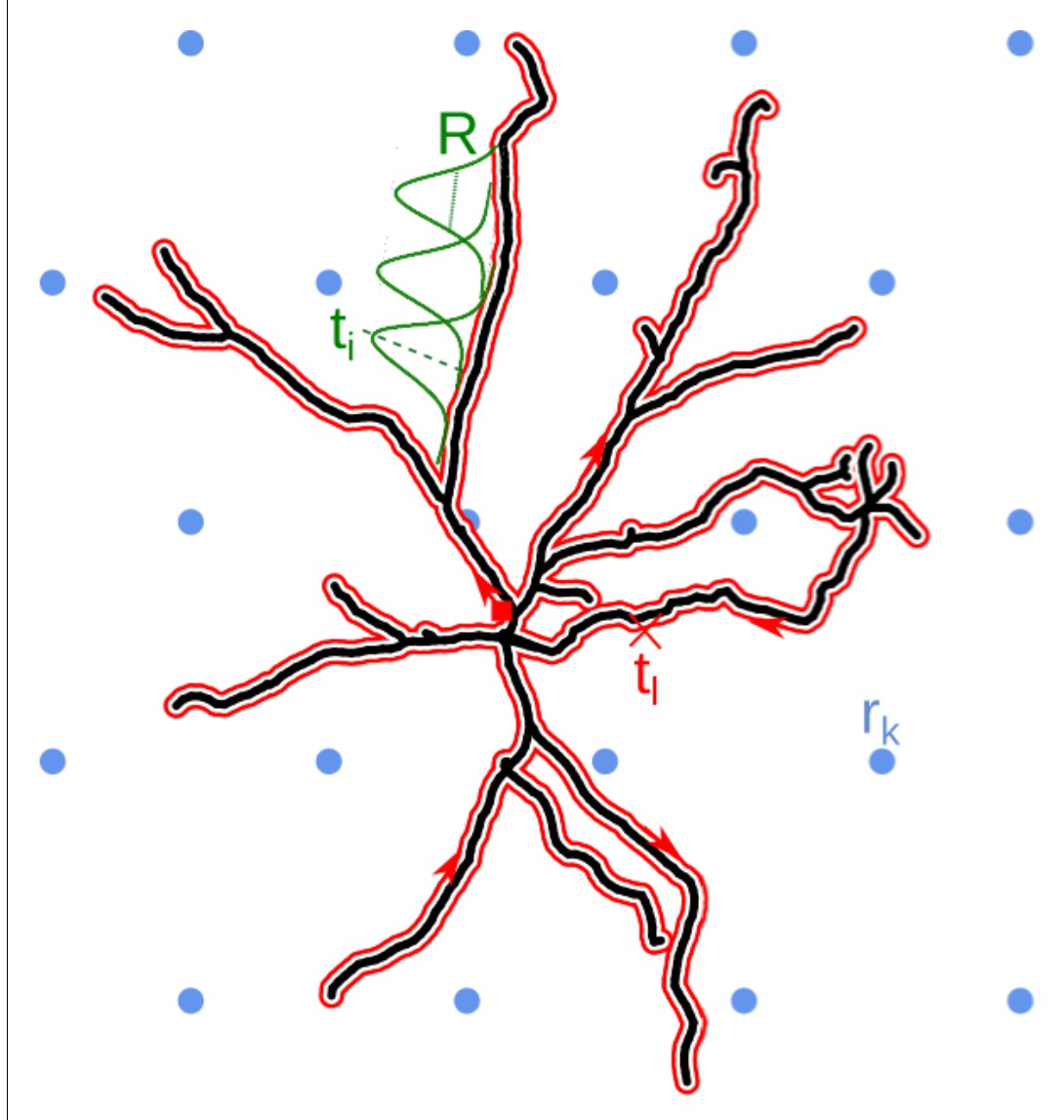

**Figure 1.** Schematic overview of the skCSD method. The black line indicates the two-dimensional projection of the neuron on the MEA plane, the blue circles mark the location of multielectrode array (hexagonal grid, in this example), $r_k$ is the position of the $k^{th}$ electrode. The morphology in our method is described by a self-closing curve in three dimensions, which is indicated by red on the plot. We shall refer to this curve as the *morphology loop*. A point of the cell is visited once, if it is a terminal point of a dendrite, more than twice, if it is a branching point and twice in all the other cases. With this strategy, any point on the morphology loop uniquely identifies the physical location of the corresponding part of the cell unambiguously. To set up estimation framework, we distribute one-dimensional, overlapping Gaussian basis functions spanning the current sources. Several of these Gaussians are plotted in green, $t_i$ marks the center of the $i^{th}$ basis element, $R$ is the width parameter.

DOI: https://doi.org/10.7554/eLife.29384.002

passing a given dendrite twice will introduce more overlap. We only require that no two functions overlay: we want them to be independent which means each two must differ, here, in practice, be shifted with respect to each other. Once the CSD basis functions are in place, we compute the potential they generate in the whole space. The distribution of the potential in space coming from a single CSD basis function is called a basis function in the potential space, or *potential basis function* for short, and is denoted by $b_j(\mathbf{x})$ (we drop the tilde).

We are now ready to start the estimation of the distribution of the current sources on the cell from the recording. This is a static procedure, in the sense that the estimated CSD at any moment in time depends only on the present value of the measurements. We are looking for the distribution of

current sources in the form of a linear combination of the basis sources which we placed. This means a weighted sum of each CSD basis source

$$C(\mathbf{x}) = \sum_{j=1}^{M} a_j \widetilde{b}_j(\mathbf{x}).$$

Then, the potential they generate, is the linear combination of the respective potential basis sources with the same weights,

$$V(\mathbf{x}) = \sum_{j=1}^{M} a_j b_j(\mathbf{x}). \tag{1}$$

Conceptually, we want to match the described potential with actual measurements and from this infer the weights. In practice, however, we cannot do it directly, because typically there will be many more basis functions, and therefore weights to be estimated, than the available measurements. To understand why this is a problem consider the simplest case, where you have two numbers, $x$ and $y$, to estimate from a single measurement which gives 1, and the physics of the problem gives equal contribution to the measurement, so that we must solve $x + y = 1$. It is easy to see that without further constraints this has an infinite number of equally good solutions.

One way to solve this problem, which was our inspiration, was proposed by *Vapnik (1998)* (Appendix to Chapter 6: Estimating functions on the basis of indirect measurements), who effectively considered the problem of estimating one quantity (here: CSD) from measurements of its function (here: potential). Here, we combine it with the kernel trick (*Schölkopf and Smola, 2002*), which allows us to make indirect estimation in the high-dimensional space of basis functions through computations in the space of measurements. We construct a *kernel function* which is a sum of products of the potential basis functions with themselves

$$K(\mathbf{x}, \mathbf{x}') = \sum_{j=1}^{M} b_j(\mathbf{x}) b_j(\mathbf{x}').$$

This function, which takes two spatial arguments, can be understood as a similarity measure between the potentials at the two points. It is easy to see, that any model of potential we can construct from our potential basis functions can also be written as a linear combination of these kernel functions with one of the variables fixed, if we use sufficiently many kernels (large $L$)

$$V(\mathbf{x}) = \sum_{l=1}^{L} \beta_l K(\mathbf{x}_l, \mathbf{x}). \tag{2}$$

The reason is that $K(\mathbf{x}_l, \mathbf{x})$ is a linear combination of all the basis functions spanning the potential space: $K(\mathbf{x}_l, \mathbf{x}) = \sum_{j=1}^{M} b_j(\mathbf{x}_l) b_j(\mathbf{x})$. Thus, if we take as many kernels as we have basis sources ($L = M$) and equate *Equation 2* with *Equation 1*, we have to solve $M = L$ equations of the form

$$a_j = \sum_{l=1}^{L} \beta_l b_j(\mathbf{x}_l)$$

for $\beta_l$, or in other words, we have to find such points $\mathbf{x}_l$ for which the above equation is solvable. This can be done, this is another way of saying that the functions $b_j(\mathbf{x})$ form a basis. It is thus fair to say that the kernel functions $K(\mathbf{x}_l, \mathbf{x})$ and $b_j(\mathbf{x})$ are equivalent basis. At this stage, it is not clear if one basis should be better than the other.

The rationale for using the kernels is provided by the Representer Theorem (*Schölkopf and Smola, 2002*), which shows that in the form of *Equation 2* we can minimize the prediction error (sum of the squared differences between the predictions of our model and actual measurements) uniquely. Moreover, the solution obtained has as many parameters as there are measurements, and we take $\mathbf{x}_l$ to be the measurement points. This is the advantage of the kernel approach over direct estimation of underlying model: here the number of parameters to be calculated is the same as the number of measurements $N$, much less than the original number of basis sources $M$, so it can be done.

Since we can expect our measurements to be noisy, a perfect fit will typically misrepresent the true potential, this is called overfitting. To counter this effect, we add a so-called *regularization* term to our error function to be minimized, which is the sum of squared parameters,

$$\sum_{k=1}^{N}(V(\mathbf{x}_k) - V_k)^2 + \lambda \sum_{l=1}^{N}\beta_l^2.$$

Thus, we want to simultaneously minimize the difference between our prediction and actual measurements while moderating the fluctuations of the interpolated potential in space. This only slightly changes the computation while significantly improving the stability of the solution. The result is

$$\beta = (\mathbf{K} + \lambda\mathbf{I})^{-1} \cdot \mathbf{V}.$$

where $\mathbf{V}$ is the vector of the measurements $V_k$, I the identity matrix, $\mathbf{K}_{jk} = K(\mathbf{x}_j, \mathbf{x}_k)$, and $\lambda$ the regularization parameter, which needs to be set.

We still need to obtain the CSD profile from the interpolated potential. In fact, we can do this without resorting directly to the basis functions. Replacing one of the potential basis functions in the definition of our kernel with the corresponding CSD basis function, we obtain what we call a *cross-kernel function*,

$$\widetilde{K}(\mathbf{x}, \mathbf{x}') = \sum_{j=1}^{M} b_j(\mathbf{x})\widetilde{b}_j(\mathbf{x}').$$

Defining

$$\widetilde{\mathbf{K}}^T(\mathbf{x}) := [\widetilde{K}(\mathbf{x}_1, \mathbf{x}), \dots, \widetilde{K}(\mathbf{x}_N, \mathbf{x})],$$

the estimated CSD is

$$C^*(\mathbf{x}) = \widetilde{\mathbf{K}}^T(\mathbf{x}) \cdot (\mathbf{K} + \lambda\mathbf{I})^{-1} \cdot \mathbf{V}.$$

The skCSD is a model-based analysis method since the specific model of CSD distribution we use, collecting of the CSD basis functions along with the model relating these functions to the measurement of potential, influences the result. This is advantageous, since all the assumptions are explicit and the user can see how they affect the result. All the estimation methods of any quantity contain assumptions, which in many cases are implicit and thus it is difficult to analyze how they affect the estimation. With all the parameters explicit we can study how their specific values affect the quality of solution. In particular, we wish to select parameters leading to the optimal solution. We do this using *cross-validation* which we shall now explain.

We select a set of parameters: $R$, $N$, $\lambda$, which fixes the model. Then, going contact by contact, we ignore the signal recorded at that particular site and build models from the remaining signals. This model predicts the potential at the ignored contact. The difference between the prediction and the actual measurement is a measure of prediction quality for a given set of parameters. We then add squares of the differences between the actual measurements at every electrode and predictions from the respective models built from all signals except the reference. Scanning through a range of parameters we look for a minimum prediction error. We use the parameters minimizing the prediction error in the subsequent analysis.

## Ball-and-stick neuron

Here, we consider the simplest neuron morphology, the so-called ball-and-stick model, which stands for the soma and a single dendrite. A virtual linear electrode was placed in parallel to the model cell 50 μm away, the electrodes were distributed evenly along the electrode extending for 600 μm.

## Increasing the density and number of electrodes improves spatial resolution of the method

To study the spatial resolution of the skCSD method, we consider the ground truth membrane current source density distributions in the form of waves with increasing spatial frequencies

$$CSD(x) = A\cos(2\pi f x/L),$$

where $A = 0.15\,nA/\mu m$ is the amplitude, $f \in \{0.5, 1, 1.5, \ldots, 12.5\}$ is the spatial frequency, $x$ is the position along the cell, $L$ is the length of the cell. Then, we compute the generated extracellular potential at the electrode locations. The laminar shank consisting of 8, 16 and 128 electrodes was placed 50 μm from the cell in parallel to the dendrite. Finite sampling of the extracellular space sets a limit to the spatial resolution of this method. Increasing the density of electrodes within the studied region leads to higher spatial precision. As shown in *Figure 2*, with 128 electrodes it is possible to reconstruct higher frequency distributions as compared to eight electrodes. This is reminiscent of the sampling theorem (*Oppenheim et al., 1997*), except here we measure the potential and reconstruct current sources, while in the sampling theorem we consider reconstruction of a continuous signal from discrete samples. What we observe is quite intuitive and typically observed in different discrete inverse methods (*Hansen, 2010*). Note that once we move to complex morphologies and random rather than regular electrode placement, the intuition we build here, that denser probing gives better spatial resolution, would hold true, even if the relation to the sampling theorem would be less apparent.

## Reconstruction of random synaptic activation

Using the ball-and-stick neuron, we now place 100 synapses along the dendrite and stimulate them randomly in time. We simulate 70 ms of recordings from this synaptically activated cell. The stimulation is sufficiently strong to evoke spiking, see Materials and methods for details. The spiking is indicated by strong red spots in the lowest first two segments in *Figure 3*, which correspond to the soma. As can be seen, the reconstructed CSD distribution reflects the ground-truth, and the precision of reconstruction improves with an increasing number of contacts, which is reflected in the reduction of cross-validation error. Notice how the reconstructed synaptic activity gets more precise with increased density of probing the potential. In particular observe how the width of the recovered synaptic activations and the somatic activations shrink with an increasing number of electrodes, which clearly shows improved resolution. This is consistent with our observations for the Fourier mode CSD profiles above. Not much change is seen in time, which is a consequence of the fact that skCSD, like all the CSD estimation methods, acts locally in time. That is, for every moment in time, the collection of potentials at this time, is analyzed. There is no direct relation to the past or future of the measured signals.

## Simple branching morphology

Let us now study the effect of branching and breaking of rotational symmetry of the cell using the skCSD method. We consider here a simple *Y*-shaped model neuron with one branching point (*Figure 4B*). We place two synapses, one on each branch (at segments 33 and 62, close to the branching point, see *Figure 4D* and *Figure 5C*). We consider both simultaneous and independent activation of these synapses, specifically, the first synapse was activated at 5, 45, 60 ms of the 70 ms long simulation, while the other was stimulated at 5, 25, 60 ms from the stimulation onset. Our goals were to determine if it was possible to separate the synaptic inputs located on two different branches, what happens at the branching point, how the arrangement of the electrodes-cell setup influences the reconstruction. We also wanted to determine if this method provides more detail about the current distribution on the cell than what is accessible from the interpolated potential and the CSD reconstructed with kCSD under the assumption of a smooth distribution of sources in space, which is the natural approach to try (*Frey et al., 2009*).

## Differentiation of synaptic inputs located on different branches

To investigate the differentiation power of the proposed approach, we consider two placements of the cell with respect to the electrode grid. Plane *xy*, in which the cell is placed in parallel to the plane of electrodes 50 μm above (*Figure 5A*), and plane *xz*, where the cell is perpendicular to the grid,

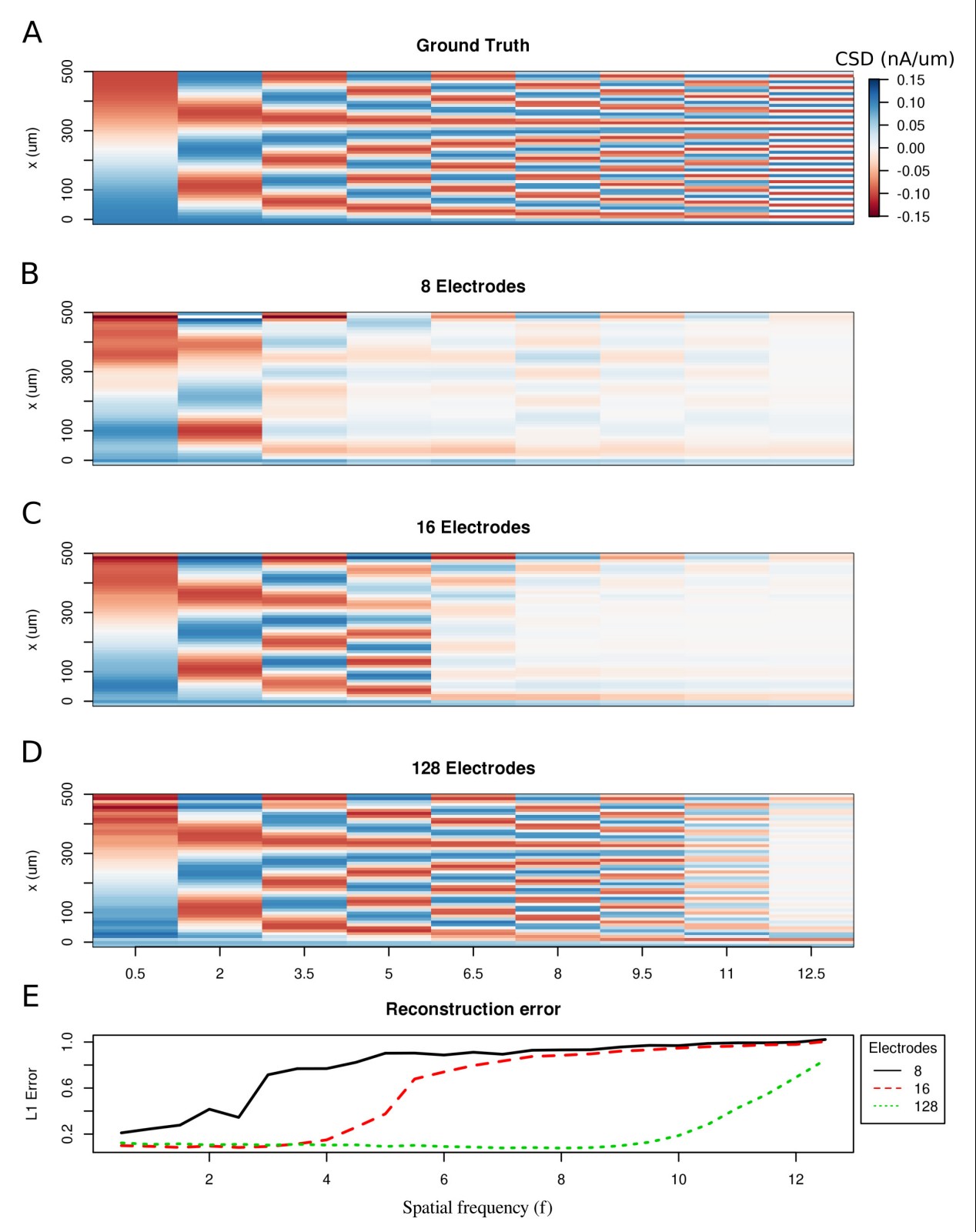

**Figure 2.** Limitations of the spatial resolution of the skCSD method in a simple ball-and-stick and laminar electrode setup. (**A**) The ground truth membrane current source density distribution was constructed from cosine waves of increasing spatial frequency (x-axis) along the cell mophology (y-axis), which is shown in the interval representation. (**B–D**) skCSD reconstruction from 8, 16 and 128 electrodes. (**E**) The L1 Error of the skCSD reconstruction for 8 (black), 16 (red) and 128 (green) electrodes for CSD patterns of increasing frequency.

*Figure 2 continued on next page*

*Figure 2 continued*

DOI: https://doi.org/10.7554/eLife.29384.003

with the grid 50 $\mu m$ away from the dendritic shaft stemming from the soma, (*Figure 5B,C*). In *Figure 5*, each panel (A–C) shows the spline-interpolated extracellular potential (V), followed by standard kCSD reconstruction, both at the plane of the 4 × 16 electrode grid used for simulated measurement. Then, the ground truth and skCSD reconstruction are shown in the branching morphology representation in the plane containing the cell morphology. Each figure is superimposed with the morphology of the cell. The dark gray shapes are guides for the eye and are sums of circles placed along the morphology with a radius proportional to the amplitude of the sources located at the center of the circle. Panel A shows results for a synaptic input depolarizing one branch. Panel B shows the same current distribution as in the previous setup, but the cell is rotated by 90 degrees with respect to the grid. In panel C synaptic input is added to the other branch. Observe that in all three cases the interpolated potential and the standard CSD reconstruction, which can be drawn only in the plane of the electrode grid, do not differ significantly, hence they cannot distinguish

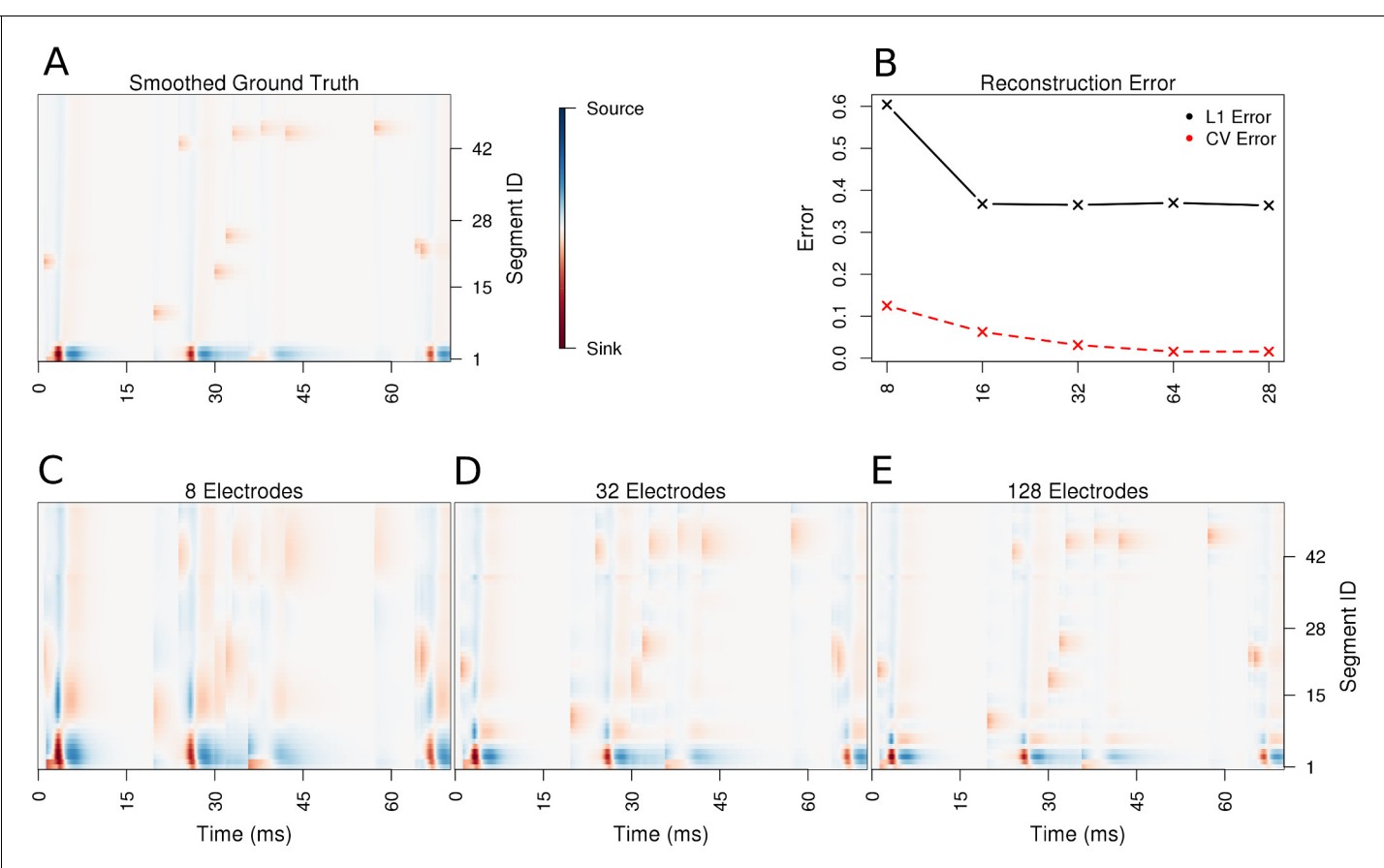

**Figure 3.** Performance of the skCSD method for a ball-and-stick neuron with random synaptic stimulation for recordings with a laminar probe placed 50 $\mu m$ away from the cell. (**A**) The ground truth spatio-temporal membrane current density in time (x-axis) along the cell in the interval representation (y-axis). The lowest segment is the soma, where the visible high amplitude of potential is a consequence of spiking. To make the much less pronounced synaptic activity on the dendritic part visible, nonlinear color map was used. Panel (**B**) shows the lowest values of cross-validation and L1 error for the before-mentioned setups. Panels (**C–E**) present the best skCSD reconstruction in case of recording with 8, 32, and 128 electrodes. One can see how increasing the number and density of probes in the region improves the reconstruction quality until a certain level. CV error was used here to select the parameters leading to the best reconstructions.

DOI: https://doi.org/10.7554/eLife.29384.004

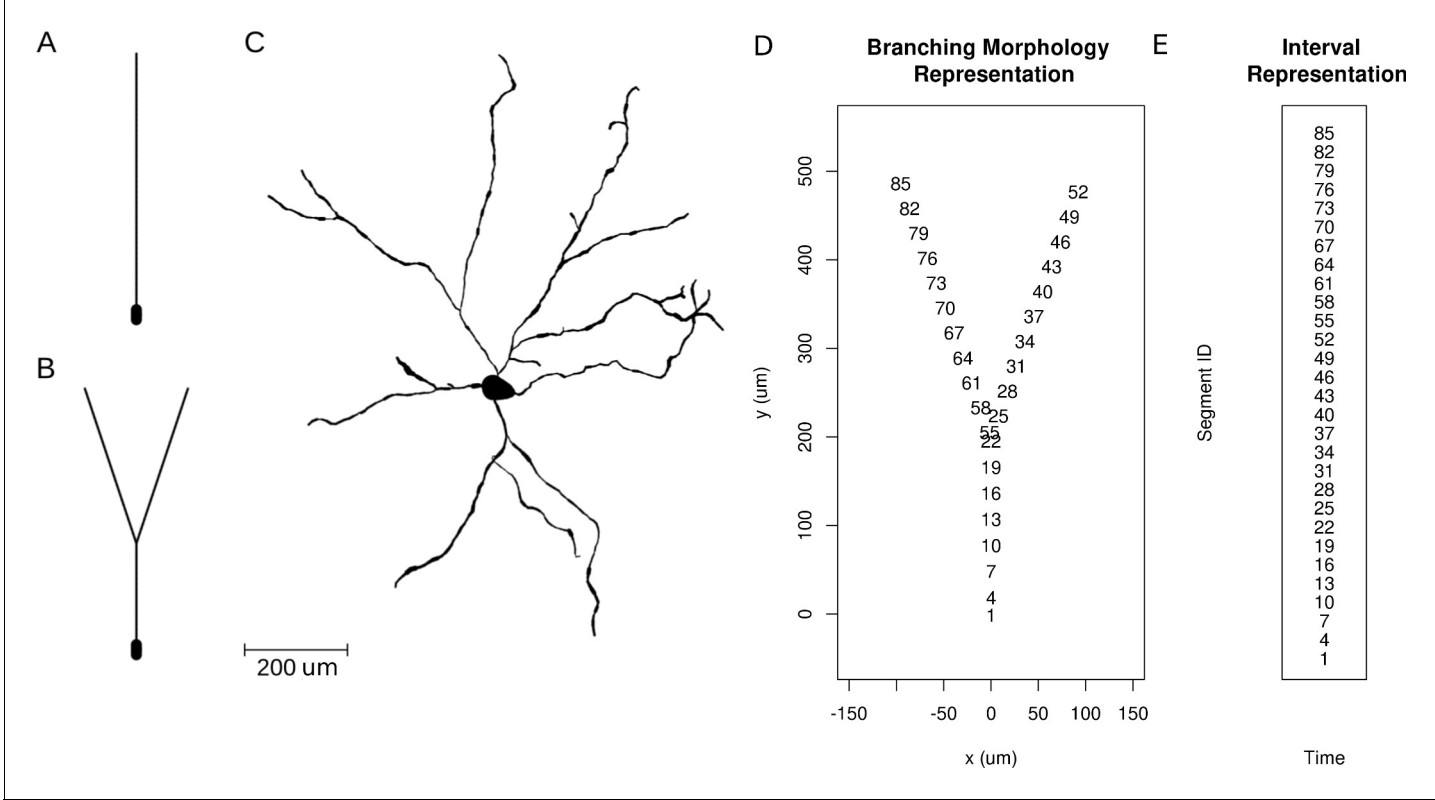

**Figure 4.** Neuron morphologies used for simulation of ground truth data. (**A**) Ball-and-stick neuron. (**B**) Y-shaped neuron. (**C**) Ganglion cell.
DOI: https://doi.org/10.7554/eLife.29384.005

between these three situations. On the other hand, the skCSD method correctly identified the synaptic inputs in all three cases.

Note that without the method proposed here, the most natural approach to analyze current sources is through use of the regular, population CSD. This approach was used, for example, to investigate the changing distribution of current sources during action potential generation using data from a high-definition MEA (**Frey et al., 2009**). What we show in figures here and below, is that while CSD (kCSD) and skCSD are consistent, using the additional information about morphology renders significantly more detail about the activity studied.

## The effect of electrode placement on skCSD reconstruction for Y-shaped cell

In **Figure 6**, we show how the number and specific distribution of the electrodes affect the quality of the reconstruction in the case of simultaneous stimulation. Panel 5. A shows the ground truth data, that is the actual distribution of the transmembrane current sources, along the morphology. To visualize it simply, we used the interval representation, the soma is shown first, followed by one branch, followed by the other. **Figure 6B** shows the reconstruction results for regularly arranged 8 (4 × 2), 16 (4 × 4), 32 (4 × 8), and 64 (4 × 16) electrodes. In **Figure 6C**, we show reconstructions for five different random placements of the same number of electrodes as for the regular case. As expected, the skCSD method is able to recover the synaptic activations and the reconstruction resolution increases with the number of electrodes. Note that in certain cases, the random distribution is more efficient than the regular grid, which is probably due to more fortunate samplings of the area covered by the morphology.

## The effect of basis on skCSD reconstruction for the Y-shaped cell

To investigate reconstruction quality in the parameter space set by the number of basis functions ($M$), basis function width ($R$) and regularization parameter ($\lambda$), we used the simulation setup for the

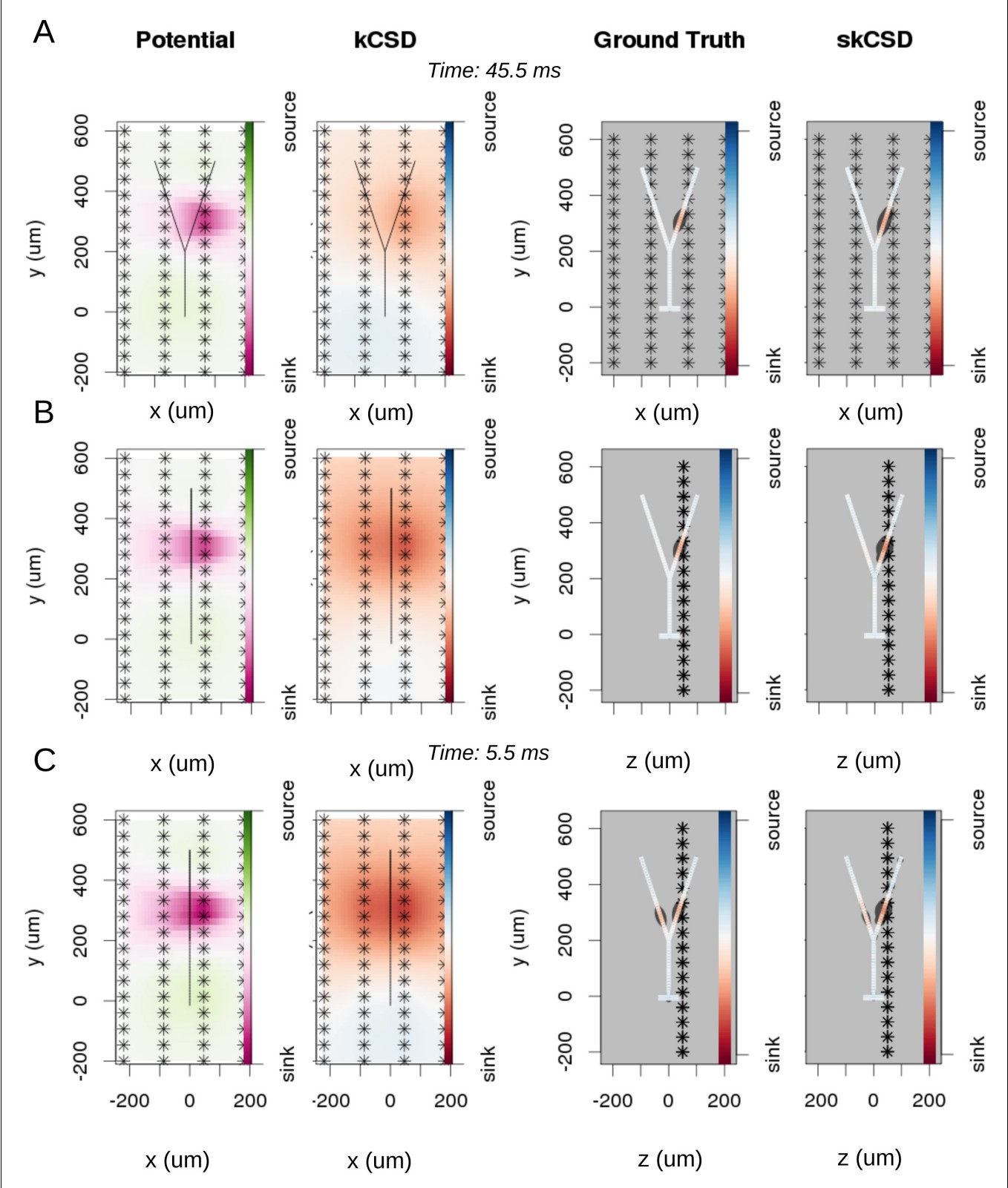

**Figure 5.** Reconstruction of synaptic inputs on a Y-shaped neuron with a regular rectangular 4 × 16 electrode grid. Each panel (**A–C**) shows the spline-interpolated extracellular potential (V), followed by standard kCSD reconstruction, both at the plane of the 4 × 16 electrodes' grid used for simulated measurement. Then, the ground truth and skCSD reconstruction are shown in the branching morphology representation in the plane containing the cell morphology. Each figure shows superimposed morphology of the cell. Note that in panel A the grid is parallel to the cell, while in panels (**B–C**) it is

*Figure 5 continued on next page*

*Figure 5 continued*

perpendicular. The dark gray shapes are guides for the eye and are sums of circles placed along the morphology with radius proportional to the amplitude of the sources at the center of the circle. (**A**) Shows results for a synaptic input depolarizing one branch. (**B**) Shows the same current distribution as in the previous setup, but the grid is rotated by 90 degrees. (**C**) A synaptic input is added to the other branch. Observe that in all three cases, the interpolated potential and the standard CSD reconstruction, which can be drawn only in the plane of the electrodes' grid, do not differ significantly, hence they cannot distinguish between these three situations. On the other hand, skCSD method is able to identify correctly both synaptic inputs.

DOI: https://doi.org/10.7554/eLife.29384.006

Y-shaped morphology with 4 × 4 electrodes. *Figure 7* shows the L1 reconstruction error for $M = 32, 128, 512, 1024$, $R = 8, 16, 32, 64, 128$, and $\lambda = 10^{-5}, 10^{-4}, 10^{-3}, 10^{-2}, 10^{-1}$. As we can see, for the smallest basis ($M = 32$) and small $\lambda$, the minimum error is obtained for wide basis sources, so that the basis functions have substantial overlap. This is necessary for the method to be able to reconstruct the family of test sources we considered. As the basis size increases, the reconstruction improves overall with minimum error obtained for narrow basis sources and small $\lambda$. The fact that we have two comparable minima for $M = 32$, for small and large $\lambda$ (top and bottom right of the plot for $M = 32$), means that the error we obtain by emphasizing the measurements (small $\lambda$) is comparable to the error we obtain by emphasizing the regularization term, which prevents over-fitting (large $\lambda$) and in effect, reflects our doubt about precision of measurement. We interpret it here as the effect of insufficient basis size. This effect disappears with increasing basis size when a unique minimum appears for moderate values of $\lambda$ and for narrow basis sources, which can best resolve small details of the CSD to be reconstructed.

## Reconstruction of current distribution on complex morphology

In this section, we consider the performance of the skCSD method in the case of a complicated, biologically realistic scenario. To achieve good spatial resolution, permitting detailed study of a cell with substantial extent, densely packed electrode arrays are required. In the present reconstruction we assumed a hexagonal grid arrangement with 17.5 μm inter-electrode distance inspired by recent experiments on reconstructing axonal action potential propagation (*Bakkum et al., 2013*; *Frey et al., 2009*). We assumed a grid consisting of 936 contacts from which we used 128 for reconstruction to be consistent with the hardware of (*Bakkum et al., 2013*; *Frey et al., 2009*).

In the simulation we assumed an experimentally plausible scenario, where oscillatory current was injected to the soma of a neuron in a slice with other inputs impinging through a 100 excitatory synapses distributed on the dendritic tree. The simulated data consisted of two parts. During the first 400 ms, the cell was stimulated by the injected current as well as through the synapses. The amplitude of the injected current was 3.6 nA, the frequency of the current drive was around 6.5 Hz. During the second 400 ms the cell was stimulated only with the current. *Figure 8* shows an example of the skCSD reconstruction at a time selected right after a spike was elicited by the cell. As we can see, neither the standard CSD reconstruction assuming smooth current distribution in space, nor the interpolated potential, give justice to the actual current distribution. At the same time, the skCSD reconstruction is quite a faithful reproduction of the ground truth. A movie comparing the ground truth with kCSD, interpolated potential, and skCSD reconstruction, in time, is provided as a supplementary material (*Video 1*).

## Dependence of reconstruction on noise level

So far, we have assumed that the data are noise-free which is never true in an experiment. Both the measurement device and the neural tissue are potential sources of distorted data. To investigate how the performance of the method is influenced by noise, we added Gaussian white noise of differing amplitudes to the simulated extracellular recordings of Y-shaped cell described above. *Figure 9A* shows the smoothed ground truth we used. The Y-shaped neuron is placed on top of a MEA with a regular grid of 4 × 8 electrodes marked by asterisks. *Figure 9B* shows the noise-free reconstruction. Panel C–F of the figure show the reconstruction results for increasing measurement noise with signal to noise ratio, SNR= 16, 4, 1. The signal-to-noise ratio (SNR) here is the standard deviation of the simulated extracellular potentials normalized with the std of the added noise. The

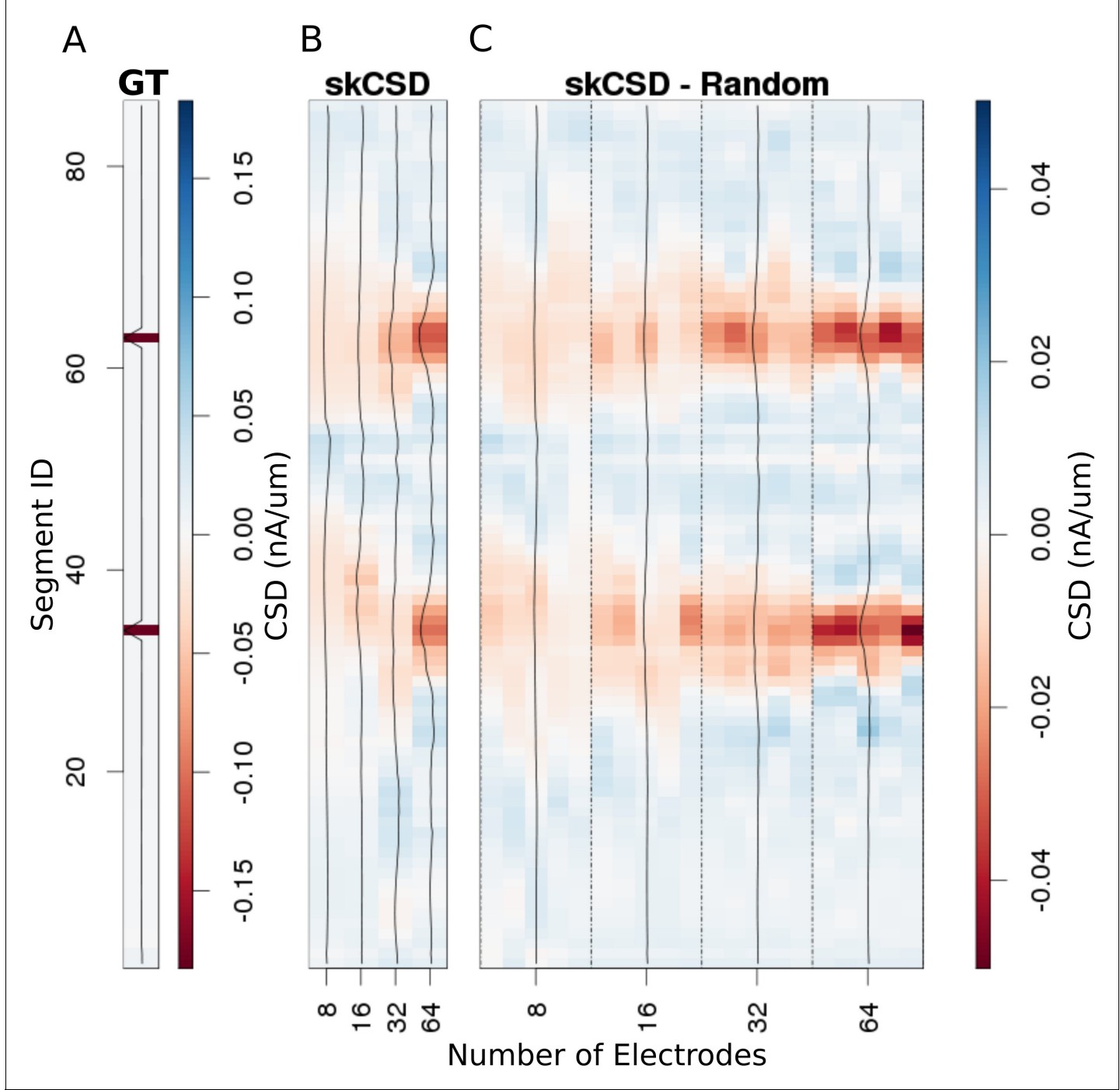

**Figure 6.** Reconstruction of synaptic inputs placed on different branches of the Y-shaped neuron for electrodes arranged regularly and randomly within the same area. We use the interval representation for visualization. The numbers on horizontal axis enumerate different electrode setups. The black profiles show the averaged membrane current as reconstructed in a given case; for random electrode distribution these are averages over five different realizations. (A) Ground truth membrane currents, the strong red indicates the synaptic inputs. (B) Reconstruction results for 8 (4 × 2), 16 (4 × 4), 32 (4 × 8), and 64 (4 × 16) electrodes arranged regularly. The skCSD reconstruction improves with the number of electrodes as the color representation and the black profiles indicate. (C) When distributing the same numbers of electrodes on the same plane as in the previous case, the quality of the average skCSD reconstruction, as indicated by the black profiles, is similar.

DOI: https://doi.org/10.7554/eLife.29384.007

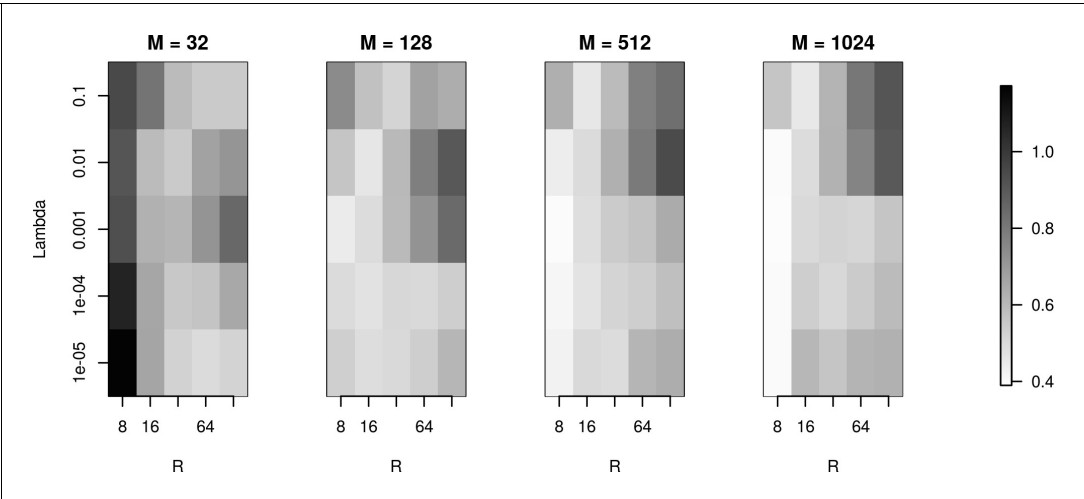

**Figure 7.** The effects of basis properties on reconstruction quality. We used the Y-shaped morphology and the 4 × 4 electrode setup to investigate the effect of using various basis numbers for the reconstruction. L1 error was calculated to compare the results for basis with $M = 32, 128, 512, 1024$ elements, for several values of basis width $R$ and $\lambda$. With few basis sources one cannot recover CSD properly. As the number of basis functions increases, the reconstruction error is minimized for moderate values of $\lambda$ and for narrow basis sources, which can best resolve small details of the CSD to be reconstructed.

DOI: https://doi.org/10.7554/eLife.29384.008

degradation of reconstruction visible in this figures is summarized in *Figure 9C*. As can be seen in the reconstruction plots (*Figure 9D–F*), the increasing noise actually does not seem to significantly alter the obtained reconstructions so the regularization is providing adequate correction, except for the noise on the order of signal (*Figure 9F*).

## Dependence of reconstruction on the number and arrangement of recording electrodes

Reconstruction of the distribution of the current sources along the morphology with skCSD (just like the reconstructions of smooth population distributions with kCSD) formally can be attempted from an arbitrary set of recordings, even a single electrode. While we do not expect enlightening results at this extreme, it is natural to ask the following questions: (1) to what extent can we trust the reconstruction in a given case, (2) which of the reconstructed features are real and which are artifacts of the method, and (3) how can the optimal parameters be selected for this method. We will return to these issues in the Discussion. Here, we wish to investigate how the number of electrodes, the density of the grid, and the area covered by the MEA, affect the results.

To answer these questions, we selected a snapshot of simulation of the model of the ganglion cell described in the Materials and methods section, with the specific membrane current distribution shown in *Figure 10A*. In *Figure 10B–H*. we show seven different reconstructions assuming different experimental setups, with differing numbers of electrodes, covering different area.

In each case, we selected the width of basis functions and the regularization parameter for the method by minimizing the L1 error calculated for the first 1000 time steps of the simulation or cross-validation error (L1-T and CV in *Figure 10I*). To verify the quality of reconstruction we computed the L1 error between the ground truth and reconstruction for the remaining 5800 time steps of the simulation. We found that minimization of L1 error gave better results and L1-V in *Figure 10I* shows the results for this case; however, the results obtained with minimization of CV error were often not much worse (not shown).

Given that L1 error can only be used where the ground truth is known, which is in simulations, we propose the following. Given the data necessary for application of the skCSD method, (neuronal morphology, positions of electrode contacts, and recorded signals) different CSD distributions should be assumed for the obtained morphology, reconstructions obtained for a range of parameters, then the L1 error could be used for optimization. Note that it is not necessary to actually

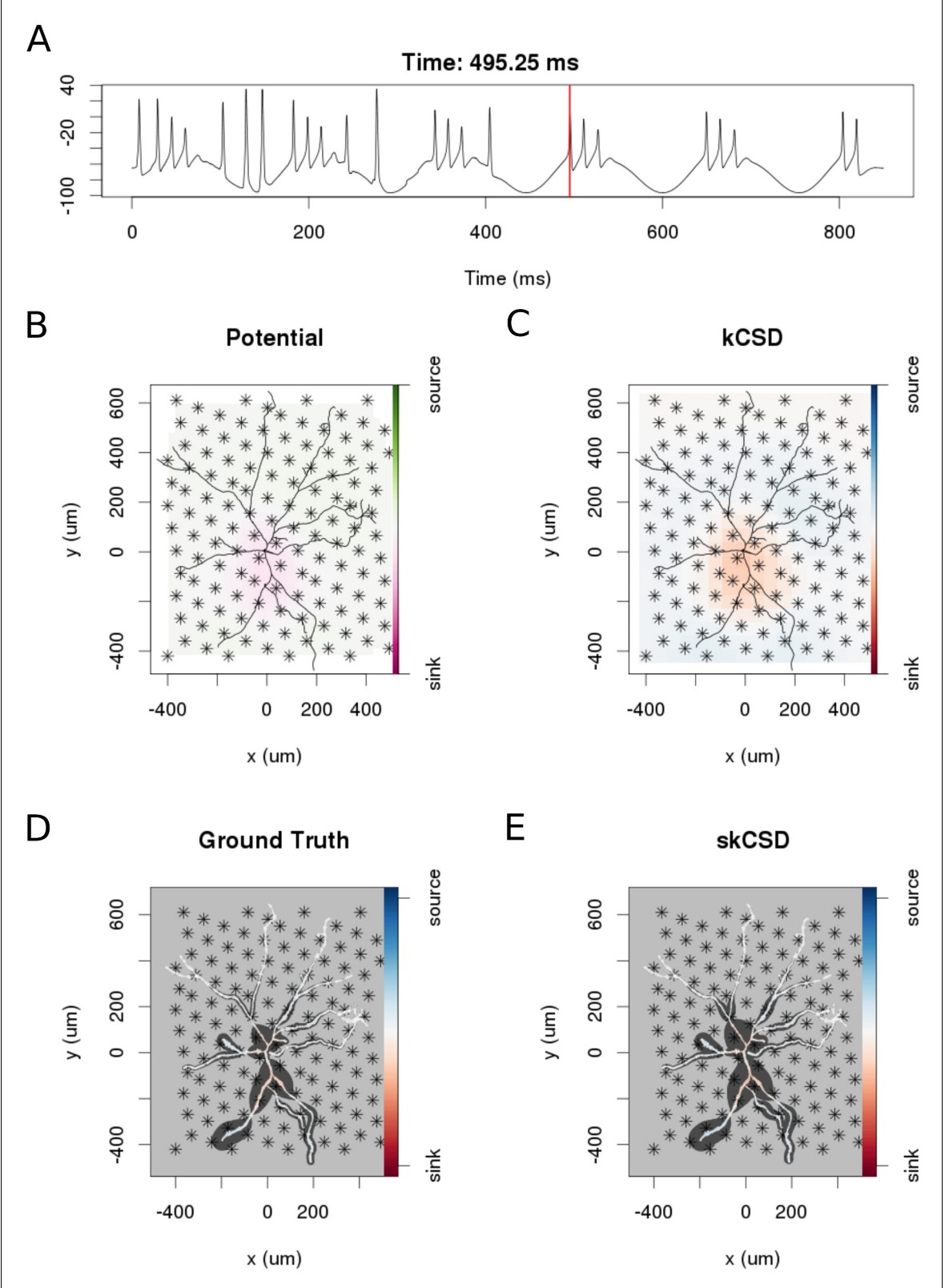

**Figure 8.** skCSD reconstruction of dendritic backpropagation patterns for a retinal ganglion cell model driven with oscillatory current. (A) Somatic membrane potential during the simulation. The red line marks the time instant for which the remaining plots were made. (B) Extracellular potential interpolated between the simulated measurements computed at the electrodes, which are marked with asterisks. (C) kCSD reconstruction computed from the simulated measurements of the potential. (D) Spatial smoothing with a Gaussian kernel was applied to the ground truth membrane current to

*Figure 8 continued on next page*

*Figure 8 continued*

facilitate comparison with the skCSD reconstruction with the same spatial resolution level. (E) skCSD reconstruction computed from the simulated measurements of the potential.

DOI: https://doi.org/10.7554/eLife.29384.009

simulate a model of the cell with proper membrane biophysics, which often is not known, although it might lead to more physiological test sources. It is sufficient to distribute the different sources along the morphology without making any assumptions concerning the biophysical properties of the neuronal membrane.

Once the parameters are obtained with this procedure, perform the analysis of actual experimental data with the obtained parameters. Performing the simulations and comparing the best reconstructions with the assumed ground truth has the further benefit of building intuition about which features of the real CSD survive in the reconstruction and which are distorted. This is another example of model-based data analysis which we believe becomes inevitable with the growing complexity of experimental paradigms, such as the one considered here.

We feel that the above procedure is optimal, since it not only gives optimal parameters, but also allows one to investigate which features are recovered and which are misformed. However, if only parameters for estimation are needed, CV error could be used, which is simpler and the results are often comparable.

The results obtained in this study are consistent with our expectations: the quality of reconstruction improves with the coverage of the morphology by the electrodes, with increasing density of probing, and with increasing number of probes (*Figure 10I*). Interestingly, it seems, that it is difficult to improve the reconstruction beyond a certain level, in consequence, the setups with moderate densities (on the order of 200 μm IED) can easily compete with setups at the edge of current developments (40 μm IED, [(*Berdondini et al., 2005*). We believe that this is not a hard limit and that better results can be obtained here. This, however, requires further development of the methods.

## Proof-of-concept experiment: spatial current source distribution of spike-triggered averages

To examine the experimental feasibility of the skCSD method, we analyzed data from a patch clamp electrode and a linear probe with 14 working electrodes recording signals simultaneously from a hippocampal pyramidal cell in an in vitro slice preparation (see Materials and methods). As there is no ground truth data available in this case, the optimal width of the basis functions and the regularization parameter were selected using the L1 error and simulated data. To do this, we used the same simulation protocol as for the ganglion cell model. A snapshot of the reconstruction is shown in *Figure 11* at the moment of firing. A 10 ms long video of the spike triggered average is shown in the supplementary materials (*Video 2*). At −0.05 ms the brief appearance of a sink (red) in the basal dendrites is visible which can be a consequence of the activation of voltage sensitive channels in the axon hillock, or the first axonal segment leading to the firing of the cell. Since there were no electrodes close to the axon initial segment, the skCSD method did not resolve it, instead it resolved to introduce the activity into the basal dendrite. This phenomenon is quickly replaced by a sink at the soma and in the proximal part of the apical dendritic tree, accompanied by sources (blue) in the basal and in the more distal apical dendrites. The extracellular potential on the second electrode reaches its minimum at 0.45 ms, which signals the peak of the spike. The deep red of the soma at this point signifies a strong sink, while the blue of the surrounding parts of the proximal apical and basal dendrites indicate the current sources set by the return currents. At 1.30 ms a source appears at the soma region, which indicates hyperpolarizing currents. Overall, the observed spatio-temporal CSD dynamics is dominated mostly by the somatic currents, responsible for the spike generation, and the corresponding counter currents. This example demonstrates the experimental feasibility of the skCSD method and may help in planning further experiments, aiming to reveal the spatial distribution and temporal dynamics of the synaptic input currents which evoke the firing of the neuron.

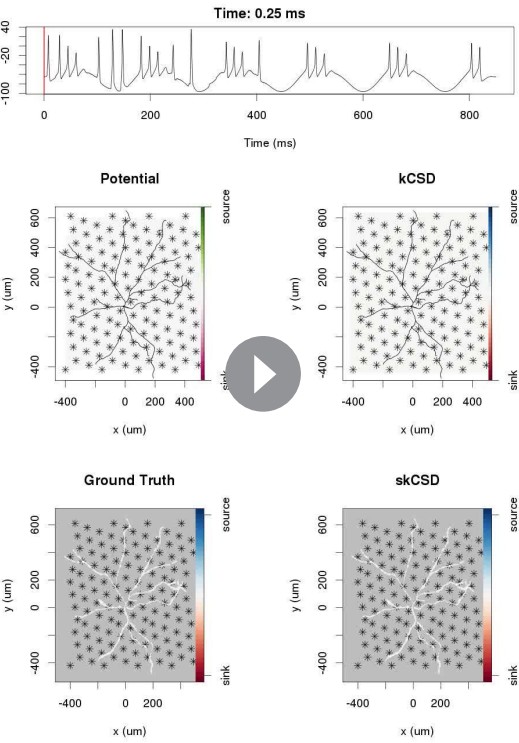

**Video 1.** skCSD reconstruction of current source density distribution on the ganglion cell. The video shows the skCSD reconstruction for the retinal ganglion cell model driven with oscillatory current (Section *Reconstruction of current distribution on complex morphology*) for the whole duration of simulation. Figure 7 shows a snapshot taken at $t = 495.25$ ms from the simulation onset. During the first 400 ms of simulation, apart from somatic drive, 100 excitatory synaptic inputs were randomly distributed along the dendrites. For reconstruction, 128 virtual electrodes were selected from the 936 arranged in a hexagonal grid of 17.5 μm interelectrode distance to record the extracellular potentials. Panel A presents the somatic membrane potential during the simulation. The red line marks the time instant for which the remaining plots were made. The colormap on Panel B shows the extracellular potential interpolated between the simulated measurements computed at the electrodes, which are marked with asterisks. The regular CSD is shown on Panel C, while the spatially smoothed ground truth membrane current is presented on Panel D. Panel E shows the skCSD reconstruction of current source density along the cell morphology from the selected measurements. The dark gray shapes are guides for the eye and are sums of circles placed along the morphology with radius proportional to the amplitude of the sources at the center of the circle.
DOI: https://doi.org/10.7554/eLife.29384.010

## Discussion

### Summary

In this work, we introduced a method to estimate the distribution of current sources (CSD) along the dendritic tree of a neuron given its known morphology and a set of simultaneous extracellular recordings of potential generated predominantly by this cell. First, assuming the ball-and-stick neuron model and a laminar probe parallel to the cell, we studied the basic viability of this method. We showed that introducing more electrodes to cover the same area leads to increased spatial resolution of the method allowing reconstruction of higher Fourier modes of the CSD generating the measured potentials (*Figure 2*). In a dynamic scenario of multiple synaptic inputs impinging on the cell, higher density of probes leads to higher reconstruction precision allowing us to distinguish individual inputs (*Figure 3*). Testing the reconstruction against the known CSD (the ground truth) shows a clear transition from poor to faithful reconstruction when the electrode distribution becomes dense enough to capture the fine detail of the CSD profile to be reconstructed (*Figure 2. E*). We also applied this method to more complex neuron morphologies, namely the Y-shape and ganglion cell. As expected, the reconstructed CSD profiles became more detailed with increasing electrode number over a fixed area (*Figures 6,10*).

Using the Y-shaped morphology we showed that (i) synaptic inputs activating different dendrites can be separated, *Figure 5*; (ii) skCSD provides meaningful information about the membrane CSD in cases, when the interpolated LFP and standard, population CSD analyses, are not informative, *Figure 5*; (iii) the reconstruction is not sensitive to a specific selection of electrode placement, *Figures 6,9*; and (iv) even significant additive noise (SNR = 1) is not prohibitive for the reconstruction, *Figure 9*.

Biologically, the most relevant example we considered was a ganglion cell model which we studied with virtual multi-electrode arrays of different designs. The MEAs we considered differed with inter-electrode distances for the simulated setups, as well as in the area they covered, ranging from an area close to the soma to roughly four times the size of the smallest square covering the whole morphology. The best results were obtained when we used the electrodes from the region which closely covered the cell (9.G and H); a reduction of inter-electrode distance from 100 μm to 40 μm yielded less impressive results than selecting the electrodes from the smallest

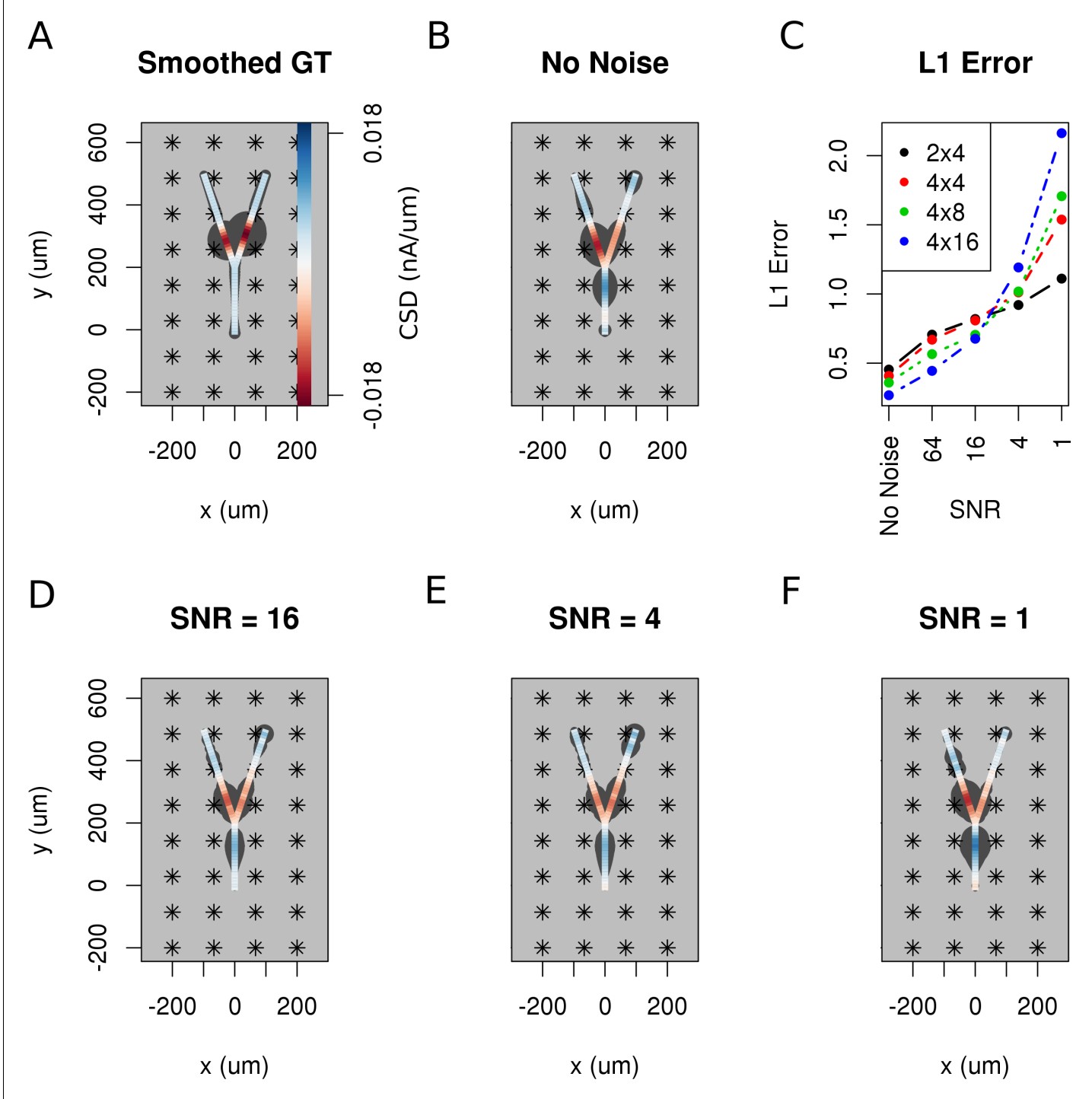

**Figure 9.** The effect of noise on the reconstruction. The corrupting influence of noise on the skCSD reconstruction is shown with the example of simultaneous excitation of both branches of the Y-shaped cell close to the branching point in case of the 4 × 8 electrodes setup. (**A**) Smoothed ground truth CSD shown on the branching morphology used. (**B,D,E,F**) skCSD reconstructions in cases of no added noise and signal-to-noise ratio equal to 16, 4, 1, respectively. Even the highest noise considered does not fully disrupt the reconstructed source distribution, although increasing the noise systematically degrades the result. This is shown in C, where the L1 error of the reconstruction was calculated for the full length of the simulations. This is consistent for different electrode setups which are marked with various colors. While the setups consisting of more electrodes perform better for low noise, the reconstruction seems to be more sensitive to noise in these cases. This might be a side effect of a specific definition of error.
DOI: https://doi.org/10.7554/eLife.29384.011

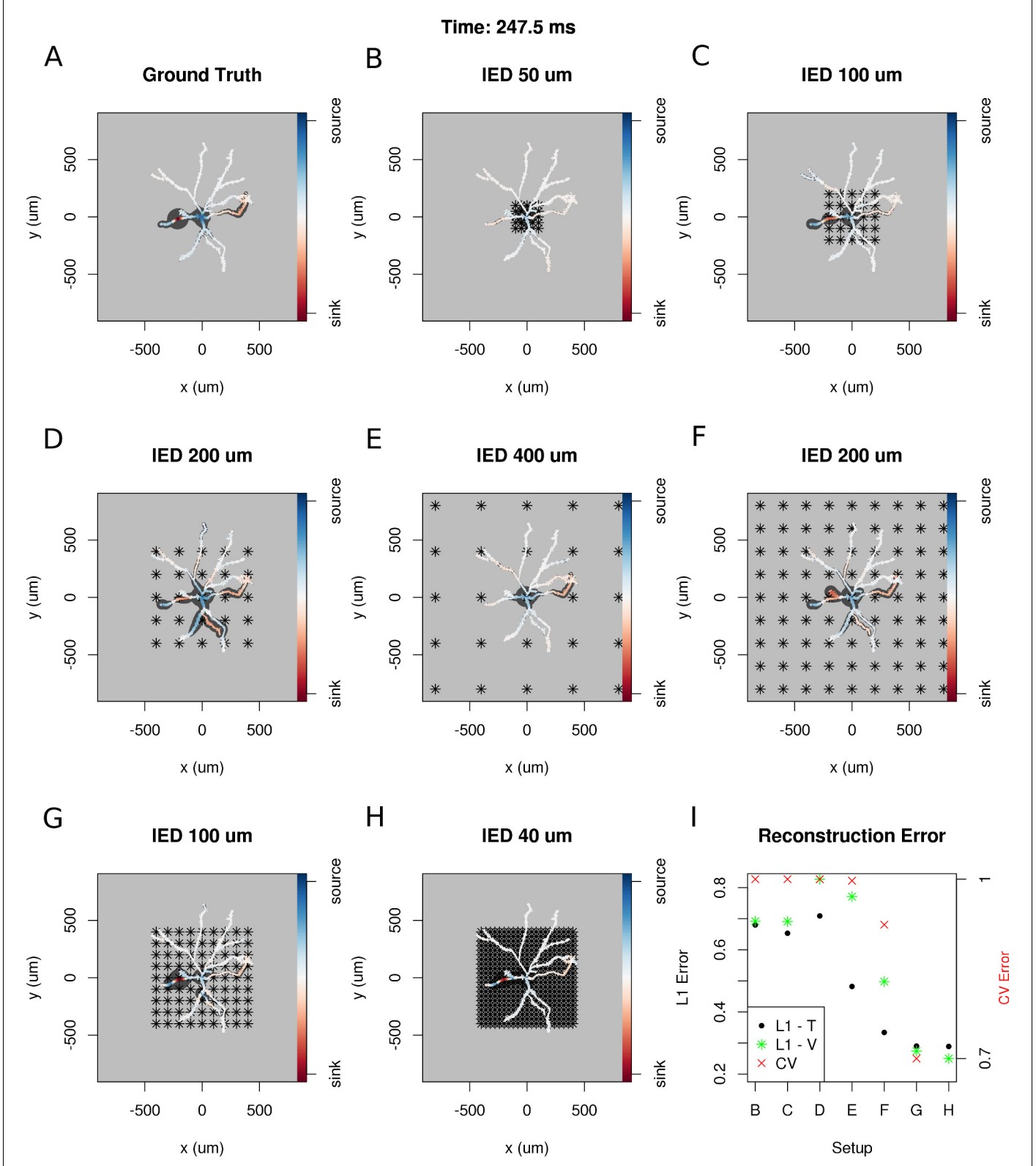

**Figure 10.** Dependence of skCSD reconstruction on multielectrode setup. Figures (**A–H**) show morphology used in the simulation together with the distribution of current sources in branching morphology representation taken at 247.5 ms of the simulation. Figures (**B–H**) show additionally the electrode setup assumed. (**A**) Smoothed ground truth CSD. (**B**) Reconstructed sources for a setup of 5 × 5 electrodes with 50 μm interelectrode distance (IED) covering a small part of the cell morphology around the soma. (**C**) Reconstructed sources for a setup of 5 × 5 electrodes with 100 μm IED

*Figure 10 continued on next page*

*Figure 10 continued*

covering a substantial part of the dendritic tree, which improves the reconstruction of the synaptic input on the left. (D) Reconstructed sources for 5 × 5 setup with 200 μm IED setup; both sinks in the membrane currents are visible. (E) Expanding the 5 × 5 electrode setup to 400 μm IED leads to a small number of electrodes placed in the vicinity of the cell which leads to a poor reconstruction. (F) Increasing the number of electrodes to 9 × 9 while keeping the coverage, which leads to 200 μm IED, does not improve the reconstruction. (G) Reducing IED in the previous example to 100 μm, which reduces the coverage of the MEA to the whole cell (same area as in panel D) bringing majority of the electrodes close to one of the dendrites, leads to one of the most faithful reconstructions among the ones shown in this figure. (H) Shows results for a matrix of 21 × 21 contacts with 40 μm IED, covering the same area as in examples D and G. The results are very good but the improvement in reconstruction does not justify the use of so many contacts with so high density. (I) Comparison of reconstruction errors for all the cases shown. Left axis: L1 error for the training (L1–T) and validation (L1–V) part . Right axis: crossvalidation error (CV). The L1-T error is marked with black points, L1-V error is represented by green stars. Generally, the L1-V errors are a bit higher than the L1-T errors but show a similar tendency. Also the CV errors, which are drawn with red crosses, show a similar tendency. The reconstructions in panels (B–H) are for parameters determined with the L1-T error.

DOI: https://doi.org/10.7554/eLife.29384.012

square covering the morphology. Our study, assumed realistic cell morphology of the ganglion cell and commercially available MEA designs, as well as realistic cell activity, showed that it is feasible to reconstruct the distribution of the current sources in realistic, noisy situations.

The skCSD method performed adequately for the proof-of-concept experimental data, even if the nature of the experiment allowed only the reconstruction of the general features of the spike-triggered average spatio-temporal current source density distribution patterns.

While the traditional (population) CSD method was used to analyze membrane currents of single cells before (*Buzsáki and Kandel, 1998*; *Bereshpolova et al., 2007*; *Frey et al., 2009*), the first CSD method specific for investigating membrane currents on single cells was proposed by (*Somogyvári et al., 2005*). However, it assumed simplified, linear neuron morphologies. An important preprocessing step proposed there was separating the single neuron's contributions to the extracellular potentials from the background activity. The novelty of the skCSD method proposed here is in its use of actual neuronal morphologies and in the underlying algorithmic solutions based on the kCSD method (*Potworowski et al., 2012*) which were initially used in the study of populations of neurons.

## General comments

Observe that skCSD, like any other data analysis method, is not a magic wand. Technically, it can be applied to data coming from a single electrode just like the age profile of a human population can be estimated from a single specimen. Obviously, in both cases, the estimate would be a poor reflection of the distribution of interest. As we improve the sampling, the quality of the estimate improves, yet ultimately it is hard to judge a priori how many electrodes is enough and if our results obtain the required level of precision. We see two approaches to address this type of questions. One is through simulations, as we discussed. The other is analysis of the singular vectors arising in the decomposition of the matrix translating the measured potentials into the estimated CSD (*Hansen, 2010*); however, the necessary tools for kCSD and skCSD are still under development. We plan to investigate this further in the future.

Having obtained the distribution of currents it would be interesting to decompose it into physiologically meaningful components, such as synaptic currents, leak currents, voltage-gated currents for different channels, etc. This seems rather challenging and we do not see a direct way of achieving this from experimental data. It is possible that an application of statistical decomposition methods will prove useful, as in the case of kCSD for population activity (*Łęski et al., 2010*; *Głąbska et al., 2014*). However, we find the contributions to the extracellular potential from individual currents highly counter-intuitive (*Głąbska et al., 2017*).

## Experimental recommendations

To attempt experimental application of skCSD we must have (1) an identified cell of known morphology, and (2) a set of simultaneous extracellular recordings of electric potential generated by this cell. Each aspect poses its challenges, some of which have been addressed here. Once we have the necessary data the natural question is how to select the parameters of the method in the specific context of a given setup, specific morphology, and recordings. Our investigations above give some

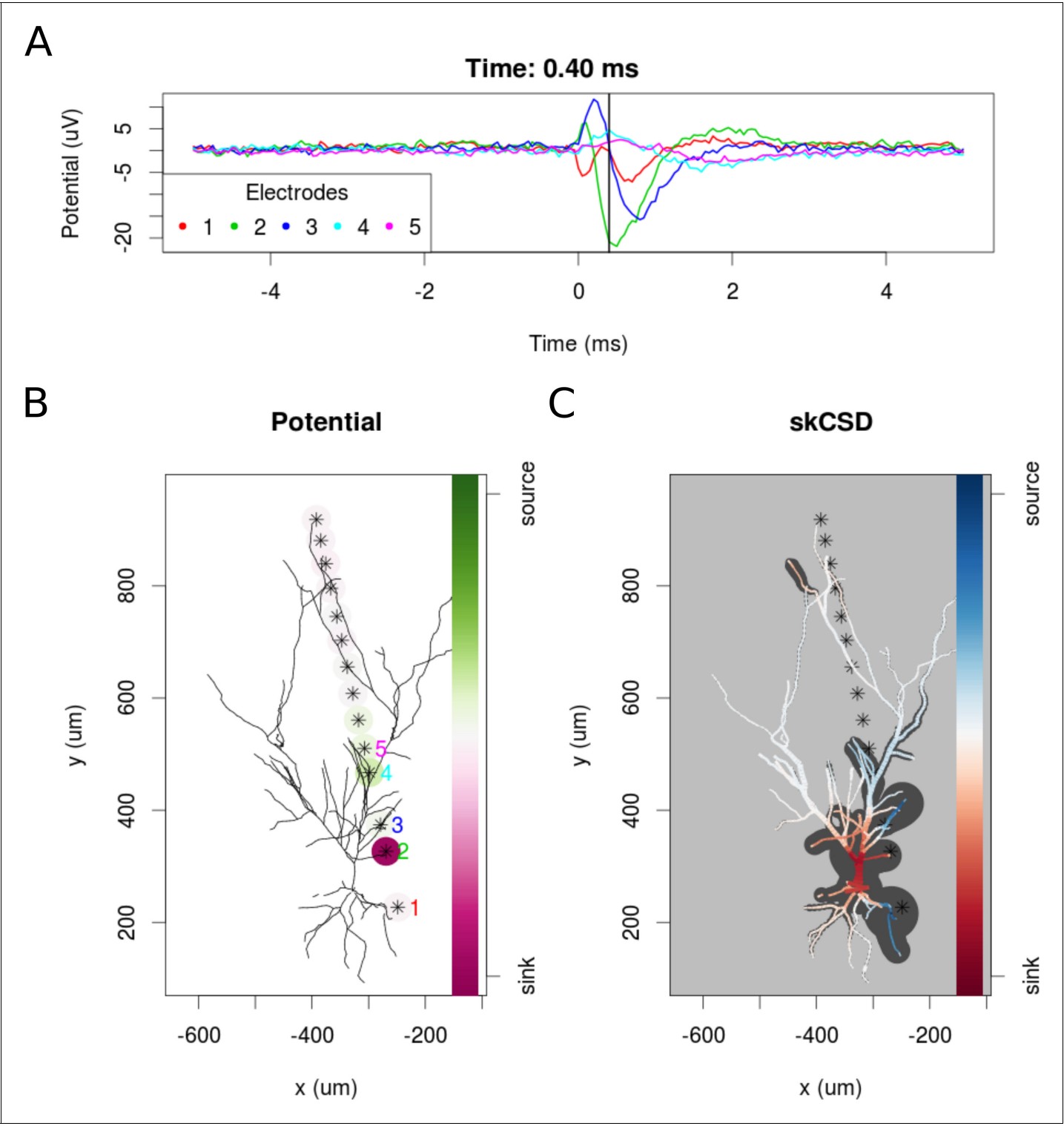

**Figure 11.** skCSD reconstruction of spike-triggered average for a hippocampal pyramidal cell (**A**) Extracellular potentials measured with the five electrodes closest to the soma. The 0 s marks the time of the membrane potential crossing the 0 mV threshold. The black vertical line marks the 0.40 ms time instant for which the extracellular potentials and skCSD reconstruction are shown. (**B**) Two-dimensional projection of the cell morphology and extracellular electrodes' positions marked by stars, the five electrodes used in the top panel of the figure are labeled with matching colors. The amplitudes of the measured potentials are shown as color-coded circles around the electrodes. (**C**) The skCSD reconstruction on the branching morphology representation. This is a snapshot of the cell firing, the red color indicates the sinks close to the soma, the blue marks the current sources on the dendrites.

DOI: https://doi.org/10.7554/eLife.29384.013

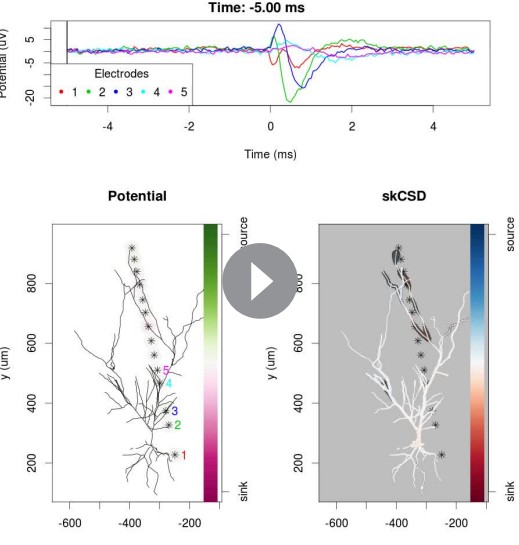

**Video 2.** Spike triggered average of pyramidal cell in vitro. The video shows the recorded potentials and skCSD reconstruction for a 10 ms time window centered around the spike as described in Section *Proof-of-Concept experiment: Spatial Current Source Distribution of Spike-triggered Averages*. The top panel presents the spike triggered averages of the potentials during 5 s before and after the spike recorded at five electrodes closest to the soma. The lower left panel shows the morphology of the cell, electrode positions, and the recorded potentials. The electrodes are marked by stars and the amplitude of the recorded potential is shown as color-coded circles around the electrodes. The snapshot is taken at the time given in the figure title and indicated by the black vertical line in the top panel. The reconstructed skCSD distribution at the same moment is shown in the lower right panel. At -0.05 ms a sink appears at the basal dendrites. This can be a consequence of the activation of voltage-sensitive channels in the axon hillock or the first axonal segment leading to the firing of the cell. Since there were no electrodes close to the axon initial segment, the skCSD method did not resolve it, instead it resolved to introduce the activity into the basal dendrite. This phenomenon is quickly replaced by a sink at the soma and in the proximal part of the apical dendritic tree, accompanied by sources (blue) in the basal and in the more distal apical dendrites. The extracellular potential on the second electrode reaches its minimum at 0.45 ms, which signals the peak of the spike. The deep red of the soma at this point signifies a strong sink, while the blue of the surrounding parts of the proximal apical and basal dendrites indicates the current sources set by the return currents. At 1.30 ms a source appears at the soma region, which indicates hyperpolarizing currents.

DOI: https://doi.org/10.7554/eLife.29384.014

indications. First, the electrodes selected for analysis should essentially uniformly cover the area spanned by the cell. This is illustrated in *Figure 6*, which shows that some degree of irregularity does not significantly affect the reconstruction. Secondly, the basis should be selected so that the basis sources could resolve the features on the membrane we are interested in (narrow basis functions) with sufficient multiplicity, that smooth coverage of the cell can be ensured (see *Figure 7*).

Clearly, as the irregularity of the setup grows we expect growing reconstruction errors. This can be studied with singular vectors of the operator transforming potential measurements into reconstructed sources, *Equation 21*, as discussed for example by (*Hansen, 2010*). We are convinced, however, that the most efficient approach to investigate the effects of these different parameters is through simulations. This is a natural place to apply the *model-based validation of data analysis* (*Denker et al., 2014*). Our suggestion is to build a computational model of the cell. We believe that for the purpose of parameter selection assuming passive membrane in the dendrites should be sufficient, but of course, more realistic biophysical information may be included, especially if available. The model cell may be stimulated with synaptic input, with current injected, or even specific profiles of ground truth CSD may be placed along the cell. Then the extracellular potential must be computed at points where the actual electrodes are placed in the experiment. One can then investigate the effects of different parameter values on reconstruction and, for the analysis of actual experimental data, select those parameters minimizing prediction error on test data. The advantage of this procedure is two-fold. First, we end up with a selection of parameters adapted for the specific problem at hand. Secondly, we build intuition regarding the interpretation of the results for our specific cell and setup. This approach is the only way to address arbitrary electrode-cell configurations and to determine how much information we can extract in a given case.

We found that the best way to identify optimal parameters for reconstruction is by minimizing the L1 error between the reconstruction and the ground truth. Since we cannot have the ground truth in an experiment, but we can assume it in the model-based validation, this is another argument for the model-based validation approach. Obviously, to efficiently apply this technique, the appropriate computational tools must be available. We plan to develop and open framework

facilitating such studies, meanwhile, the code used for the present study is available at https://github.com/csdori/skCSD (*Cserpán, 2017*; copy archived at https://github.com/elifesciences-publications/skCSD).

### Challenges of recording extracellular potentials and obtaining the morphology from the same cell

Although recording extracellular potential with a MEA, filling a neuron with a dye, and reconstructing its morphology, are standard experimental techniques, using them simultaneously remains a challenge due to the size of the experimental devices which need to be arranged within a small volume. Cells in the vicinity of the MEA can be filled individually by intracellular or juxtacellular electrodes, or with bulk dying. Individual recording and dying with a glass electrode provides not only the morphology, but also unambiguous spike times, giving an opportunity to determine the extracellular potential fingerprint of the recorded cell on the MEA. Although these would be favorable data, intracellular recording less than 100 µm from the MEA is extremely challenging. Experimental setups featuring the necessary equipment already exist (*Neto et al., 2016*), but as far as we know, have not been used in this way. On the other hand, bulk dying techniques result in more filled neurons, although the quality of the dying, and thus the quality of the 3D morphology reconstructions, is considerably lower in these cases. Although there are methods for estimation of the cell position relative to the MEA (*Somogyvári et al., 2005*, *Somogyvári et al., 2012*), association of multiple optically labeled neurons with the recorded extracellular spike patterns is still unsolved.

### Challenges of separating the activity of a single neuron from background

We propose two experimental scenarios one could apply to separate the activity of the studied neuron from the background. If we can sort the spikes elicited by the neuron of interest we can calculate the spike-triggered averages of the potentials reducing all uncorrelated contributions. Unfortunately, in live tissue, contributions from neighboring cells will have some correlations due to shared inputs. Separation of the contribution of the neuron of interest from the correlated background can be obtained in two ways. One is decomposition of the activity into meaningful components, for example, as shown by (*Somogyvári et al., 2015*), the high amplitude correlated oscillatory background of hippocampal theta activity can be extracted with independent component analysis, allowing the determination of cell-type-specific time course of the synaptic input. Alternatively one could consider combining skCSD with population kCSD analysis, that is, consider basis sources covering the cell of interest as well as the space covering the whole population. This will be the subject of further study. The second experimental scenario to obtain the contributions to the extracellular potential from a specific cell is to drive the cell with intracellular current injection of known pattern, for example, with an oscillatory drive as we discussed (*Figure 8*), and by averaging over multiple periods (event-based triggering). Again, further study is needed to establish the validity of this type of experimental procedure.

### Challenges of using novel MEAs

Handling data from high-density MEAs with thousands of electrodes will require further studies, as the large numbers of small singular values of the kernel matrix may introduce numerical sensitivity to the reconstruction. Also, optimal selection of electrodes in case of programmable MEAs merits further investigations. We believe it is best to address such issues when actual experiments are attempted.

### Importance of this work

Traditional electrophysiology has focused on the electrical potential, which is relatively easy to access, from intracellular recordings, all kinds of patch clamp, juxtacellular, to extracellular and voltage-sensitive dyes (*Covey and Carter, 2015*). While the relation of the actual measurement to the voltage at a point may significantly differ, often this is a reasonable interpretation, if needed, more realistic models of measurement can be considered, for example, averaging over the contact surface for extracellular electrodes, etc (*Moulin et al., 2008*; *Ness et al., 2015*).

Already in the middle of the 20th century, Walter H. Pitts observed that for recordings obtained with regular grids one can approximate the Poisson equation to estimate the distribution of current sources in the tissue (*Pitts, 1952*). His approach assumed recordings on a regular 3D grid, which was challenging to obtain for some 60 years (*Łęski et al., 2007*). However, with the work of Nicholson and Freeman (*Nicholson and Freeman, 1975*) 1D CSD analysis became attractive, as summarized by Ulla Mitzdorf (*Mitzdorf, 1985*). In 2012, we proposed how to overcome the restriction of regular grids with a kernel approach which both allows to use arbitrary distribution of contacts and corrects for noise (*Potworowski et al., 2012*). All the previous work, however, always assumed the contributions to the extracellular potential coming from the whole tissue and smooth in the estimation region.

In the present work, we show for the first time how a collection of extracellular recordings in combination with a cell morphology can be used to estimate the current sources located on the cell contributing to the recorded potential. Since it is now feasible experimentally to obtain the relevant data, we believe that the method proposed here may find its uses to constrain the biophysical properties of the neuron membrane, facilitate verification of morphological reconstructions, as well as guide new discoveries by offering a more global picture of the distribution of the currents along the cell morphology, giving a coherent view of the global synaptic bombardment and return currents within a cell.

# Materials and methods

## Key resources table

| Reagent type (species) or resource | Designation | Source or reference | Identifiers | Additional information |
|---|---|---|---|---|
| Strain, strain background (Wistar rat, male) | Male Wistar rat | PMID: 11619935 | | |
| Biological sample (Wistar rat, male) | Hippocampal slice | | | |
| Chemical compound, drug | Biocytin | PMID: 17990268 | | |
| Software, algorithm | Neurolucida Stereo Investigator | PMCID: PMC3332236 | RRID:SCR_001775 | |
| Software, algorithm | RHD2000-Series Amplifier Evaluation System Intan Technologies, LLC | Intan Technologies, http://intantech.com/aboutus.html | | |
| Software, algorithm | LFPy | doi: 10.3389/fninf.2013.00041 | RRID:SCR_014805 | |
| Software, algorithm | NEURON | https://neuron.yale.edu/neuron/ | RRID:SCR_005393 | |
| Software, algorithm | Python Programming Language | http://www.python.org | RRID:SCR_008394 | |
| Software, algorithm | R Project for Statistical Computing | https://www.r-project.org/ | RRID:SCR_001905 | |
| Software, algorithm | NeuroMorpho.Org | PMCID: PMC2655120, http://neuromorpho.org/ | RRID:SCR_002145 | |
| Software, algorithm | Kernel Current Source Density Python library | PMID: 22091662 | RRID:SCR_015777 | |
| Software, algorithm | skCSD method | this paper, https://github.com/csdori/skCSD | | A tool for estimating transmembrane currents along the dendritic tree of a neuron from extracellular recordings |
| Other | Ganglion cell morphology | PMID: 20826176, http://neuromorpho.org/neuron_info.jsp?neuron_name=Badea2011Fig2Du | | |

## Overview of current source density reconstruction methods

### Traditional CSD

For reader's convenience, here we briefly present the basic ideas behind the traditional and recent approaches to reconstruction of current source density (CSD analysis). For a more complete review of CSD analysis see *Wójcik (2015)*, for recent reviews of the relations between neural activity, current sources and the recordings see (*Buzsáki et al., 2012*; *Einevoll et al., 2013a*).

The relation between current sources in the tissue and the recording potentials is given by the Poisson equation

$$C = -\nabla(\sigma\nabla V), \tag{3}$$

where $C$ stands for CSD and $V$ for the potential. While this can be studied numerically for nontrivial conductivity profiles (*Ness et al., 2015*), here we shall mostly assume a constant and homogeneous conductivity tensor, $\sigma$. In that case, the above equation simplifies to $C = -\sigma\Delta V$ and can be solved for $C$ given potential in the whole space. On the other hand, given the CSD in the whole space, the potential is given by

$$V(\mathbf{x}) = \frac{1}{4\pi\sigma}\int d^3\mathbf{x}' \, \frac{C(\mathbf{x}')}{|\mathbf{x}-\mathbf{x}'|}. \tag{4}$$

Walter Pitts observed that having recordings on a regular grid of electrodes we can estimate CSD by taking numerical second derivative of the potential (*Pitts, 1952*), we call this approach *traditional CSD method*. Pitt's idea gained popularity only after Nicholson and Freeman popularized its use for laminar recordings (*Nicholson and Freeman, 1975*) in the cortex. In this setup, assuming the layers are infinite and homogeneous (*Pettersen et al., 2006*), the current source density at each layer can be estimated from

$$C(z_j) = -\sigma\frac{V(z_j+h) - 2V(z_j) + V(z_j-h)}{h^2}, \tag{5}$$

where $z_j$ is the position of the $j^{th}$ electrode and $h$ is the inter-electrode distance.

### Inverse CSD (iCSD)

To overcome limitations of the traditional approach, such as difficulty of handling the data at the boundary and hidden assumptions about the dimensions we do not probe, Pettersen et al. proposed a model-based *inverse CSD method* (*Pettersen et al., 2006*). Initially proposed in 1D, the method was later generalized to other dimensionalities (*Łęski et al., 2007*; *Łęski et al., 2011*). Given a set of recordings $V_1, \ldots, V_N$ at regularly placed electrodes at $\mathbf{x}_1, \ldots, \mathbf{x}_N$ this method assumes a model of CSD parametrized with CSD values at the measurement points, $C(\mathbf{x}) = \sum_{k=1}^{N} C_k f_k(\mathbf{x})$, where $f_k(\mathbf{x})$ are functions taking 1 at $\mathbf{x}_k$, 0 at other measurement points, with the values at other points defined by the specific variant of the method, for example, spline interpolated in spline iCSD (*Wójcik, 2015*). Assuming the model $C(\mathbf{x})$ one computes the potential at the electrode positions obtaining a relation between the model parameters, $C_k$, and the measured potential, $V_k$, which can be inverted leading to an estimate of the CSD in the region of interest.

### Kernel CSD (kCSD)

The kernel Current Source Density method (*Potworowski et al., 2012*) can be considered a generalization of the inverse CSD. It is a non-parametric method which allows reconstructions from arbitrarily placed electrodes and facilitates dealing with the noise. Conceptually, the method proceeds in two steps. First, one does kernel interpolation of the measured potentials. Next, one applies a 'cross-kernel' to shift the interpolated potential to the CSD. In 3D, in space of homogeneous and isotropic conductivity, this amounts to applying the Laplacian to the interpolated potential, *Equation. (3)*. To handle all cases in a general way, including data of lower dimensionality or with non-trivial conductivity, we construct the interpolating kernel and cross-kernel from a collection of basis functions. The idea is to consider current source density in the form of a linear combination of basis sources $\widetilde{b}_j(\mathbf{x})$, for example Gaussian,

$$C(\mathbf{x}) = \sum_{j=1}^{M} a_j \widetilde{b}_j(\mathbf{x}), \tag{6}$$

where the number of basis sources $M \gg N$, the number of electrodes, and $a_j$ are the weights with which the basis sources are combined into the model CSD. Let $b_j(\mathbf{x})$ be the contribution to the extracellular potential from $\widetilde{b}_j(\mathbf{x})$, which in 3D is

$$b_j(x,y,z) = \mathcal{A}\widetilde{b}_j(x,y,z) = \frac{1}{4\pi\sigma} \int dx' \int dy' \int dz' \frac{\widetilde{b}_j(x',y',z')}{\sqrt{(x-x')^2 + (y-y')^2 + (z-z')^2}}, \tag{7}$$

but in 1D or 2D we would need to take into account the directions we do not control in experiment (for example, along the slice thickness for a slice placed on a 2D MEA). Then, the potential will have a form

$$V(\mathbf{x}) = \mathcal{A}C(\mathbf{x}) = \sum_{i=1}^{M} a_i b_i(\mathbf{x}). \tag{8}$$

Since we cannot estimate $M$ coefficients $a_j$ from $N$ measurements for $N<M$, we construct a kernel for interpolation of the potential,

$$K(\mathbf{x}, \mathbf{x}') = \sum_{i=1}^{M} b_i(\mathbf{x}) b_i(\mathbf{x}'). \tag{9}$$

Then, any potential field $V(\mathbf{x})$ spanned by $b_i(\mathbf{x})$ can be written as

$$V(\mathbf{x}) = \sum_{l=1}^{L} \beta_l K(\mathbf{x}_l, \mathbf{x}), \tag{10}$$

for some $L$, $\mathbf{x}_l$, and $\beta_l$, but it minimizes the regularized prediction error

$$\sum_{k=1}^{N} (V(\mathbf{x}_k) - V_k)^2 + \lambda \sum_{l=1}^{L} \beta_l^2, \tag{11}$$

when $L = N$. Here, $\mathbf{x}_k$ are the positions of the electrodes, $V_k$ are the corresponding measurements, $\lambda$ is the regularization constant. The minimizing solution is obtained for

$$\boldsymbol{\beta} = (\mathbf{K} + \lambda\mathbf{I})^{-1} \cdot \mathbf{V}. \tag{12}$$

where $\mathbf{V}$ is the vector of the measurements $V_k$, and $\mathbf{K}_{jk} = K(\mathbf{x}_j, \mathbf{x}_k)$.

To estimate CSD we introduce a cross-kernel

$$\widetilde{K}(\mathbf{x}, \mathbf{x}') = \sum_{j=1}^{M} b_j(\mathbf{x}) \widetilde{b}_j(\mathbf{x}'). \tag{13}$$

If we define

$$\widetilde{\mathbf{K}}^T(\mathbf{x}) := [\widetilde{K}(\mathbf{x}_1, \mathbf{x}), \ldots, \widetilde{K}(\mathbf{x}_N, \mathbf{x})],$$

then the estimated CSD takes form of

$$C^*(\mathbf{x}) = \widetilde{\mathbf{K}}^T(\mathbf{x}) \cdot (\mathbf{K} + \lambda\mathbf{I}))^{-1} \cdot \mathbf{V}, \tag{14}$$

where $\lambda$ is the regularization parameter and I the identity matrix; see (*Potworowski et al., 2012*) for derivation and discussion.

## Spike CSD (sCSD)

The Spike CSD (*Somogyvári et al., 2012*) is the forerunner of the method presented here, as it aims to estimate the current source distribution of single neurons with unknown morphology. The sCSD method provides an estimation of the cell-electrode distance and uses a simplified model of the shape of the neuron to reach this. Separating potential patterns generated by different neurons is critical and it is obtained by clustering extracellular fingerprints of action potentials which are different for every neuron. The limitation of sCSD is the assumed simplified morphology of the model and low spatial resolution. Despite that, even with this simplified model, it was possible to demonstrate for the first time the EC observability of backpropagating action potentials in the basal dendrites of cortical neurons, the forward propagation preceding the action potential on the dendritic tree and the signs of the Ranvier-nodes (*Somogyvári et al., 2012*).

## skCSD method

The single-cell kCSD method (skCSD), which we introduce in this work, is an application of the kCSD framework where we assume that the measured extracellular potential comes mainly from a cell of known morphology and known spatial relation to the MEA. To estimate the CSD in this case, we must cover the morphology of the cell with a collection of basis functions. To do this, a one-dimensional parametrization of the cell morphology is needed. This could be done independently for each branch of the neuron or globally for the whole cell at once. While the first approach might seem easier, handling of the branching point is non-trivial. Instead, we decided to fit a closed curve on the morphology, which we call the *morphology loop* (*Figure 1*). This curve should cover all the segments of the cell, be as short as possible, and be aligned with the morphology. For example, in case of a ball-and-stick neuron, the curve starts at the soma, goes towards the tip of the dendrite, turns back, goes back to the soma, and closes there. One parameter $s$ is enough to unambiguously determine a position on this line, although most points on the morphology are mapped to two $s$ parameters. We also need a method to handle the branching points and guide the parametrization so that all the branches will be visited in an optimal way. This problem is a special case of the Chinese postman problem known from graph theory (*Kwan, 1962*). Given this information, we can distribute the basis functions $\widetilde{b}_j(\mathbf{x})$ along the morphology of the cell (*Figure 1*).

In practice, based on the morphology information we define an ordered sequence of all the segments such that the consecutive segments are always physically connected and preference is given to those neighbors which have not been visited yet. The process is continued until all the segments are covered and the last element in the sequence connects to the first element. Note that in the sequence the final segments of the branches are present once, the branching point multiple times and the intermediate ones twice. Then we fit a spline on the coordinates of the segments following the ordered sequence resulting in a morphology loop construction. The CSD basis functions are distributed along this loop uniformly. Any point $\mathbf{x} \equiv (x, y, z)$ on the morphology can be parameterized with $s \in [0, l]$ on the loop:

$$
\begin{aligned}
x &= f_x(s), \\
y &= f_y(s), \\
z &= f_z(s),
\end{aligned}
\tag{15}
$$

where $l$ is twice the length of all the branches. Consider the following basis functions:

$$
\widetilde{b}_i(s) = e^{-(s-s_i)^2/R^2}
\tag{16}
$$

where $s_i$ is the location of the $i$-th basis function on the morphology loop, $R$ its width.

The contribution to the extracellular potential from a basis source $\widetilde{b}_i(s)$ is given by

$$
b_i(x, y, z) = \frac{1}{4\pi\sigma} \int \frac{\widetilde{b}_i(s)}{\sqrt{(x - f_x(s))^2 + (y - f_y(s))^2 + (z - f_z(s))^2}} ds.
\tag{17}
$$

As in kCSD, for CSD of the form

$$C(s) = \sum_{i=1}^{M} a_i \widetilde{b}_i(s)$$

we obtain the extracellular potential as

$$V(\mathbf{x}) = \sum_{i=1}^{M} a_i b_i(\mathbf{x}). \qquad (18)$$

As before, for estimation of potential we use kernel interpolation. Note that in this case the basis functions in the CSD space, $\widetilde{b}(s)$, live on the morphology loop, while the basis functions in the potential space, $b_i(\mathbf{x})$, live in the physical 3D space. To determine the current source density distribution along the fitted curve, we introduce the following kernel functions:

$$K(\mathbf{x}, \mathbf{x}') = \sum_{j=1}^{M} b_j(\mathbf{x}) b_j(\mathbf{x}'), \qquad (19)$$

$$\widetilde{K}(s, \mathbf{x}') = \sum_{j=1}^{M} \widetilde{b}_j(s) b_j(\mathbf{x}'). \qquad (20)$$

With these definitions the regularized solution for $C$ on the morphology loop is given by *Equation 14*:

$$\mathbf{C}(s) = \widetilde{\mathbf{K}}^T(s)(\mathbf{K} + \lambda \mathbf{I})^{-1} \mathbf{V}. \qquad (21)$$

To obtain the distribution of currents at a given point in space we need to sum the currents on the loop at points which are mapped to that physical position $\mathbf{x}$:

$$\mathbf{C}(\mathbf{x}) = \sum_{s:r(s)=\mathbf{x}} \mathbf{C}(s). \qquad (22)$$

## Construction of ground truth data

To validate the method, we used simulated data which allows us to consider arbitrary cell-electrode setups and test various current patterns. The LFPy package (*Lindén et al., 2013*) was used to simulate the extracellular potential at arbitrarily placed virtual electrodes. We assumed the .swc morphology description format (*Cannon et al., 1998*) and the sections were further divided to segments. The coordinates of every segment's ends were used to find the connections. Once the connection matrix was calculated, we used the Chinese postman algorithm to obtain the morphology loop. We calculated the potential using neuron models with various morphologies shown in *Figure 4* and different input distributions, assuming one- and two-dimensional multielectrode arrays. We used toy models to better understand and characterize the method as well as a biologically realistic neuron model to estimate performance of skCSD in an experimentally realistic scenario.

The simplest setup we used was a ball-and-stick neuron recorded with a laminar probe. Various artificial CSD patterns and also biologically more realistic CSD distributions served as test distributions in order to quantify the spatial resolution and reconstruction errors. To generate the ground truth data we simulated a 500 µm long linear cell model of 52 segments in LFPy. The diameter of the two segments representing the soma was 20 µm, while the other segments were 4 µm wide. 100 synaptic excitation events were distributed randomly along this morphology in order to imitate a biologically realistic scenario.

To test the effect of branching on the results, a simple Y-shaped morphology was used (*Figure 4B*). The synapses were placed at segments 33 and 62 on different branches close to the branching point. The first was stimulated at 5, 45, 60 ms, the other at 5, 25, 60 ms after the onset of the simulation. The idea here was to consider the inputs stimulated together and separately. The times of activation were randomly selected in such a way as to leave enough time for the membrane activity to settle down. This can be viewed as extreme cases of correlated and uncorrelated events. Note that the skCSD reconstruction is not affected directly by the temporal correlation of the synaptic inputs. Just like any CSD estimation method, skCSD is applied to the potentials recorded at a given point in time. Of course, it is affected indirectly, in the sense that slower or faster oscillating

inputs lead to different spatial patterns due to filtering effects of the dendritic membrane: fast oscillations induce short dipoles, slow oscillations allow the current to spread along the cell leading to stronger dipoles (*Lindén et al., 2010*). We address these effects indirectly in *Figure 2*.

As a realistic example, we used a mouse retinal ganglion cell morphology (*Kong et al., 2005*) from NeuroMorpho.Org (*Ascoli, 2006*). In the simulations 608 segments were used. 100 synaptic excitation events were distributed randomly along this morphology within the first 400 ms of the simulation. The cell was also driven with an oscillatory current. In the dendrites, only passive ion channels were used.

## Parameters of the simulations

We simulated three different model morphologies: ball-and-stick (BS), Y-shaped (Y), and a ganglion cell (Gang). The Y-shaped neuron was considered in two situations, when it was parallel (Y) or orthogonal (Y-rot) to the MEA plane. The extracellular potential was computed at multiple points modeling different experimentally viable recording configurations (cell and setup). All combinations used are summarized in *Table 1*. The parameters describing the neuron membrane physiology are given in *Table 2*. The length of the simulation was 70 s in case of the ball-and-stick and Y-shaped neurons, and 850 s for the ganglion cell model.

## Parameters of synapses

In most simulations we modeled synaptic activity. We used synapses with discontinuous change in conductance at an event followed by an exponential decay with time constant τ (ExpSyn model as implemented in the NEURON simulator). When simulating the Y-shaped neuron we placed two synapses with the following parameters: reversal potential: 0 mV , synaptic time constant: 2 ms, synaptic weight: 0.04 μS. The synapses were placed at segments 33 and 62 (see *Figures 4,5*). When simulating the other models (ball-and-stick and ganglion cell) we used the same type of synapse; however, the synaptic weights were a quarter of the above (0.01 μS) since they were more numerous (*Table 2*).

## Measuring the quality of reconstruction

To validate the skCSD method, we need to consider two situations. When we know the ground truth — the actual distribution of sources which generated the measured potentials — we can compare the reconstruction with it. This is available directly only in simulations. In that case, we can measure the prediction error between the reconstruction and the original. However, the skCSD method by its nature gives smooth results. This is a consequence of kernel interpolation of the potential which occurs in the first step of the method. The same phenomenon occurs in regular CSD estimation (*Wójcik, 2015*). Thus, we can never recover the original CSD distribution but only a coarse-grained approximation. This is not a significant problem as the coarse-grained CSD should have equivalent physiological consequence. However, to compare the reconstructed density with the ground-truth, which is typically very irregular in consequence of multiple synaptic activations, we always smoothed the ground truth CSD with a Gaussian kernel. The width of the kernel was 15 μm for ball-and-stick model, while for the Y-shaped and ganglion cell models we used 30 μm.

Thus, whenever ground truth was known, we computed L1 norm of the difference between the reconstruction $C^*$ and smoothed ground truth $C$ normalized by the L1 norm of $C$:

**Table 1.** Main parameters of the simulated cells and setups.

| | Cell properties | | Synapse properties | | | Distribution of electrodes | |
|---|---|---|---|---|---|---|---|
| | Length ($\mu m$) | Number of Seg. | Location (ID of Seg.) | Number of Syn. | Synaptic Weight ($\mu S$) | Type | Number |
| **BS** | 516 | 53 | random | 100 | 0.01 | linear | 8,16, 32, 64, 128 |
| **Y** | 848 | 86 | 33, 62 | 6 | 0.04 | rectangular, random | 2 × 4, 4 × 4, 4x8, 4x16 |
| **Y-rot** | 848 | 86 | 33, 62 | 6 | 0.04 | rectangular | 8,16, 32, 64 |
| **Gang** | 5876 | 608 | random | 100 | 0.01 | hexagonal, rectangular | 128, 25, 49, 81, 441 |

DOI: https://doi.org/10.7554/eLife.29384.015

**Table 2.** Biophysical parameters characterizing the simulated cell models.

| Quantity | Value | Unit |
|---|---|---|
| Initial potential | −65 | $mV$ |
| Axial resistance | 123 | $\Omega cm$ |
| Membrane resistivity | 30000 | $\Omega cm^2$ |
| Membrane capacitance | 1 | $\mu F/cm^2$ |
| Passive mechanism reversal potential | −65 | $mV$ |

DOI: https://doi.org/10.7554/eLife.29384.016

$$\epsilon_{L1} = \frac{\sum\limits_{segments,time} |\mathbf{C} - \mathbf{C}^*|}{\sum\limits_{segments,time} |\mathbf{C}|}. \tag{23}$$

When analyzing experimental data we only have access to the noisy measurements and cannot apply the above strategy directly. Thus we consider two strategies. One is to use cross-validation error (CV). In leave-one-out cross-validation (*Potworowski et al., 2012*) we estimate CSD from all the measurements but one and compare estimated prediction with actual measurement on the removed electrode. Repeating this procedure for all the electrodes gives us a measure of prediction quality for a given set of parameters for this specific dataset. Scanning over some parameter range we identify optimal parameters as those giving minimum error. They are further used to analyze the complete data. The advantage of using cross-validation error is that it does not require the knowledge of the ground truth current source density distribution and can still provide an estimation about the performance of the skCSD method. As this algorithm is quadratic in the number of electrodes, for large arrays one might prefer to use the leave-p-out cross-validation instead. When we test how the quality of the reconstruction changes with the number of electrodes we use CV error normalized by the number of electrodes which can then be compared between different setups.

The other strategy we use and recommend in the experimental context, when we know the cell morphology and its geometric relation to the setup, as well as the measurements, is model-based analysis. The idea is to simulate different current source distributions, either placing specific distribution by hand or by modeling activity of the cell assuming passive membrane and random or specific synaptic activations, both of which are relatively inexpensive both in computational time and coding complexity. This reduces the problem to the modeling case. We can use thus generated data (CSD and potentials) scanning for optimal reconstruction parameters to be used in analysis of actual experimental data from the setup.

To handle the effects of noise one should study its properties on electrodes, for example, assuming white measurement noise identify its variance, then tune the regularization parameter $\lambda$ on simulated sets with comparable simulated noise added.

## Parameter selection

To apply the skCSD method, we need to decide upon the number of basis functions, set their width ($R$), and choose the regularization parameter $\lambda$. In this work, the number of basis function was set to 512 for all cases, which is at least twice the number of electrodes used. This is usually not a limitation, the more the better. For the basis width (*Equation. (16)*) we took the following values: 8, 16, 32, 64, 128 μm. Selection of the regularization parameter is not trivial (*Potworowski et al., 2012*; *Hansen, 2010*). Here, we tested the effect of the regularization parameter taking values of 0.00001, 0.0001, 0.001, 0.01, 0.1 The optimal parameters were identified by the lowest value of reconstruction error.

## Visual representation of CSD on the morphology

To visualize the distribution of current sources and other quantities along a neuron morphology we use two representations of the cell:

1. *Interval representation*: we stack all the compartments consecutively along the y-axis so that the part of the dendrite stemming from the soma is shown first, followed by one branch, followed by the other. The order of the branches in the stack is taken from the morphology loop to make these representations consistent. The *x*-axis either shows different time instants of the simulations or various distribution patterns.
2. *Branching morphology representation*: in this case a two-dimensional projection of the cell is shown which is colored according to the amplitudes of the membrane current source densities at a time instant. To visually enhance the current events, gray circles proportional to the amplitude of CSD at a point are placed centered at the point to facilitate comprehension.

## Experimental methods

### In vitro experiment

One male Wistar rat (300 g) was used for the slice preparation procedure. The in vitro experiment was performed according to the EC Council Directive of November 24, 1986 (86/89/EEC) and all procedures were reviewed and approved by the local ethical committee and the Hungarian Central Government Office (license number: PEI/001/695-9/2015). The animal was anesthetized with isoflurane (0.2 ml/100 g). Horizontal hippocampal slices of 500 μm thickness were cut with a vibratome (VT1200s; Leica, Nussloch, Germany). We followed our experimental procedures developed for human in vitro recordings (*Kerekes et al., 2014*), adapted to rodent tissue. Briefly, slices were transferred to a dual superfusion chamber perfused with artificial cerebrospinal fluid. Intracellular patch-clamp recordings, cell filling, visualization and three-dimensional reconstruction of the filled cell was performed as described in (*Kerekes et al., 2014*). For the extracellular local field potential recordings, we used a 16-channel linear multielectrode (A16 × 1–2 mm-50-177-A16, Neuronexus Technologies, Ann Arbor, MI), with an INTAN RHD2000 FPGA-based acquisition system (InTan Technologies, Los Angeles, CA). The system was connected to a laptop via USB 2.0. Wideband signals (0.1−7500 Hz) were recorded with a sampling frequency of 20 kHz and with 16-bit resolution. The recorded neuron was held by a constant −40 nA current injection.

### Data preprocessing

One hundred and fifty-four spikes were detected on the 180 s long intra-cellular recording by 0 mV upward threshold crossing. A ± 5 ms wide time windows were cut around the moments of each spikes on each channels of the extra-cellular (EC) potential recordings and averaged, to access the fine details of the EC spatio-temporal potential pattern which accompanied the firing of the recorded neuron on all channels. Two channels were broken (2, 5); however, as the skCSD method allows retrieving CSD maps from arbitrarily distributed contacts, this has not prevented the analysis; the broken channels were excluded from further consideration. The averaged spatio-temporal potential maps were high-pass filtered by subtracting a moving window average with 100 ms width. This filtering, together with the spike-triggered averaging procedure, ensured that the resulted EC potential map contains only the contribution from the actually recorded cell. The price we paid was filtering out EC signals of the spontaneous repetitive sharp-wave like activity of the slice which was correlated by the firing of the recorded neuron and thus the presumptive synaptic inputs of the recorded neuron as well. An additional temporal smoothing by a moving average with 0.15 ms window was used to reduce the effect of noise.

## Acknowledgements

Research supported by grants from the Polish Ministry of Science and Higher Education 2729/7.PR/2013/2, the Hungarian National Research, Development and Innovation Fund NKFIH K 113147, K 119443 and NN 118902, and the Hungarian National Brain Research Program KTIA NAP 13-1-2013-0001 and KTIA-13-NAP-A-IV/1,2,3,4,6. We are grateful for Emese Pálfi and László Négyessy for the opportunity of using their Neurolucida setup at the Department of Anatomy, Histology and Embryology, Semmelweis University, to Chaitanya Chintaluri and Mark J. Hunt for critical reading of the manuscript, and to Joanna Jędrzejewska-Szmek for help with testing and documenting the software for public exposition.

## Additional information

### Funding

| Funder | Grant reference number | Author |
|---|---|---|
| Nemzeti Kutatasi, Fejlesztesi es Innovacios Hivatal | K119443 | Lucia Wittner |
| Nemzeti Agykutatasi Program | KTIA-13-NAP-A-IV/1 2 3 4 6 | István Ulbert |
| Nemzeti Agykutatasi Program | KTIA NAP 13-1-2013-0001 | Zoltán Somogyvári |
| Nemzeti Kutatási, Fejlesztesi és Innovacios Hivatal | K 113147 | Zoltán Somogyvári |
| Nemzeti Kutatasi, Fejilesztesi es Innovacios Hivatal | NN 118902 | Zoltán Somogyvári |
| Ministerstwo Nauki i Szkolnict-wa Wyższego | 2729/7.PR/2013/2 | Daniel K Wójcik |

The funders had no role in study design, data collection and interpretation, or the decision to submit the work for publication.

### Author contributions

Dorottya Cserpán, Conceptualization, Data curation, Software, Formal analysis, Validation, Investigation, Visualization, Methodology, Writing—original draft, Writing—review and editing; Domokos Meszéna, Data curation, Investigation, Methodology; Lucia Wittner, Kinga Tóth, Investigation, Methodology; István Ulbert, Conceptualization, Supervision, Funding acquisition, Methodology; Zoltán Somogyvári, Resources, Supervision, Funding acquisition, Methodology, Writing—original draft, Writing—review and editing; Daniel K Wójcik, Conceptualization, Supervision, Funding acquisition, Investigation, Methodology, Writing—original draft, Writing—review and editing

### Author ORCIDs

Dorottya Cserpán (iD) https://orcid.org/0000-0002-7538-1931
Domokos Meszéna (iD) http://orcid.org/0000-0003-4042-2542
Lucia Wittner (iD) http://orcid.org/0000-0001-6800-0953
Kinga Tóth (iD) http://orcid.org/0000-0002-8751-8499
István Ulbert (iD) http://orcid.org/0000-0001-9941-9159
Zoltán Somogyvári (iD) http://orcid.org/0000-0002-4385-3025
Daniel K Wójcik (iD) http://orcid.org/0000-0003-0812-9872

### Ethics

Animal experimentation: The in vitro experiment was performed according to the EC Council Directive of November 24, 1986 (86/89/EEC) and all procedures were reviewed and approved by the local ethical committee and the Hungarian Central Government Office (license number: PEI/001/695-9/2015).

### Decision letter and Author response

Decision letter https://doi.org/10.7554/eLife.29384.024
Author response https://doi.org/10.7554/eLife.29384.025

## Additional files

### Supplementary files

• Transparent reporting form
DOI: https://doi.org/10.7554/eLife.29384.017

## Major datasets

The following previously published dataset was used:

| Author(s) | Year | Dataset title | Dataset URL | Database, license, and accessibility information |
|---|---|---|---|---|
| Badea TC, Nathans J | 2011 | Neuron morphology Badea2011Fig2Du | http://neuromorpho.org/neuron_info.jsp?neuron_name=Badea2011Fig2Du | Publicly available at NeuroMorpho.Org (ID : NMO_10743) |

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
