## [Decision Letter]

Thank you for submitting your article "Revealing the Distribution of Transmembrane Currents along the Dendritic Tree of a Neuron from Extracellular Recordings" for consideration by *eLife*. Your article has been favorably evaluated by Richard Aldrich (Senior Editor) and three reviewers, one of whom, Frances Skinner, is a member of our Board of Reviewing Editors. The following individuals involved in review of your submission have agreed to reveal their identity: Alexandra Chatzikalymniou (Reviewer #1), Joshua Goldwyn (Reviewer #2); Michiel Remme (Reviewer #3).

The reviewers have discussed the reviews with one another and the Reviewing Editor has drafted this decision to help you prepare a revised submission.

This paper introduces a novel method to estimate current sources underlying extracellular recorded voltages. The method relies on a combination of kernel CSD analysis of the extracellular potential recordings and knowledge of the morphology of a neuron close to the recording electrodes. Assuming one is able to isolate the contribution of this one cell to the extracellular potentials, this method can map the current sources onto this cell. The method was validated against arbitrary morphologies and electrode configurations simulating biologically relevant scenarios from simple (ball and stick and Y shape neuron) to more complicated (ganglion cell model). It is thought that this method has great potential and should be interesting to many neurophysiologists as it allows for significantly improved reconstructions of the current source density distribution along a cell morphology.

However, the reviewers noted various aspects that need to be addressed.

Essential revisions:

1) The code needs to be included and not just upon request.

Please see the information on *eLife* website regarding Tools and Resources papers: https://submit.elifesciences.org/html/*eLife*_author_instructions.html#types

"…relevant code must conform to the Open Source Definition and be deposited in an appropriate public repository; and methodological advances need to be comprehensibly described…"

2) The authors need to edit their paper in the following ways:

a) Overall editing of the paper is needed.

There are numerous spelling errors and especially the articles before many nouns are missing. Much more proofreading of the text is recommended as part of the overall revisions.

b) Bring forth the significance/importance/need for their work earlier. For example, the last section of the Discussion ('importance of this work') could be moved to the beginning of the paper.

c) Edit and expand writing to be more accessible, to be more understandable to the general reader, and to avoid confusion in what is meant.

Specifically: i) Since this paper introduces a new methodology, and most readers will not have a e.g. machine learning background, I believe the paper needs to improve the explanation of the methodology, and not just in the Materials and methods, but throughout the paper. Clearly the authors have already attempted to make it readable to non-experts, but I would specifically recommend the following:

- At the beginning of the results, a summary of the method should be included. That is, present the method in a "self-contained" way so that the reader does not need to 'walk through' several methods (kCSD, sCSD, skCSD) and remarks on the innovations in each "iteration" as given in the Materials and methods at end of the paper.

- It should be possible to read the paper from beginning to end without running into terms that have not been defined yet (e.g. subsection “Dependence of reconstruction on noise level”: 'regularization'; subsection “Dependence of reconstruction on the number and arrangement of Recording Electrodes”, third paragraph: 'the width of the basis functions'; 'the cross-validation error').

- In the Materials and methods: some intuition for the kernel method should be given (and not by referring to another paper). Why would one 'avoid direct estimation of coefficients a_j'?. What is gained with the kernel trick? One will have to estimate fewer parameters, but how does that come about? Does this rely on specific assumptions?

ii) Is it meaningful to make comparisons to interpolated voltage method or kCSD method? These methods are not, to my knowledge, designed or expected to have resolution at the single cell (or subcellular level) – well, I suppose the resolution is determined by the electrode grid, but in any case it is not clear to me that readers would assume that these methods "should" be able to resolve branch points or other information local to a dendritic tree. Edit/expand the manuscript to make clear what is meant.

iii) “Proof of concept experiment”. I am confused by this section. The authors go into detail describing the spatial and dynamical features of the inferred transmembrane currents. But then they seem to nullify their observations and the usefulness of their method when they write "From an experimental setup consisting of only 14 electrodes on a linear probe a detailed distribution of current sources along a complex morphology cannot be expected". Then, they write "but the firing activity is well observable" – but this point is not novel or contested. Indeed, spike identification from multi-electrode recordings is a common data analysis task, and not the (apparent) point of the method presented here.

See also paragraph in Discussion beginning: "The skCSD method performed adequately for the proof of concept experimental data". I don't feel this is justified, without further clarification of the message in the subsection “Proof of Concept experiment: Spatial Current Source Distribution of Spike-triggered Averages”. Edit/expand the manuscript to make clear what is meant.

iv)“To test the effect of branching on the results, a simple Y-shaped morphology was used (Figure 11). […] The first was stimulated at 5, 45, 60 ms, the other at 5, 25, 60 ms after the onset of the simulation.” Why choose these arbitrary stimulation times? Why are the synapses activated in this specific temporal sequence? It would be interesting to know what the temporal resolution of the technique is. Also, how would the performance of the technique change for different degrees of correlation of synaptic inputs? Edit/expand the manuscript to address reasoning/rationale of these questions.

v)“As a realistic example, we used a mouse retinal ganglion cell morphology Kong et al. (2005) from NeuroMorpho.Org Ascoli (2006). […] The cell was also driven with an oscillatory current. In the dendrites, only passive ion channels were used.” What is the frequency of the oscillatory input and its effect? They do say approximately 6.5Hz (line 175), but it would to helpful to know about different frequencies (say higher γ frequencies) and/or discuss it. How does the separation sensitivity of current sources depend on the frequency of the oscillatory input used? The validity of the technique should be tested for different oscillatory frequencies so that potential users keep in mind how the performance of current sources separation may vary as a function of input frequency. Is the separation of synaptic inputs equally successful for lower and higher synaptic input frequencies? As users will likely need to reveal current sources of different oscillatory input frequencies this information would be important to present. Edit/expand the manuscript to do and/or discuss these aspects.

vi) The Introduction (third paragraph) suggested to me that the current injection to the cell is exploited in this new method. However, this idea does not seem to be very central throughout the paper. Perhaps this could be stated more clearly.

3) Estimating parameters in real experiments.

The authors suggest using a modeling approach to generate data and optimize the parameters and then use these parameters for the actual data.

- First of all, from the Figure 8 results it seems the CV error is pretty good (though what is the CV error for case H?). Why not just use that? It would be informative if the authors illustrate how different the CSD estimates are when using the CV error instead of the L1 error in Figure 9.

- My main concern with using the modeling approach (L1 error) is that the active properties of the neuron are not known sufficiently well. But I would expect that the presence of active currents (e.g. sodium, calcium, NMDA spikes, but also subthreshold active currents) would have a significant impact on the parameter estimates (R, λ etc.), compared to the parameters for a passive model (see suggestion in the subsection “Experimental Recommendations”). The authors should explore this, i.e. show how parameter estimates vary when considering a passive versus some variations of active models. There are various detailed hippocampal pyramidal cell models in modelDB from which the authors could use the active properties for their cell reconstruction and compare the obtained skCSD parameters with those obtained from a passive model.

- In general, the effects of the parameters λ, R and M are not properly discussed. How strongly do they affect the CSD estimates?

---

## [Author Response]

Essential revisions:1) The code needs to be included and not just upon request.Please see the information on eLife website regarding Tools and Resources papers: https://submit.elifesciences.org/html/eLife_author_instructions.html#types"…relevant code must conform to the Open Source Definition and be deposited in an appropriate public repository; and methodological advances need to be comprehensibly described…"

We have now placed the code at https://github.com/csdori/skCSD. This is described in the manuscript.

2) The authors need to edit their paper in the following ways:a) Overall editing of the paper is needed.There are numerous spelling errors and especially the articles before many nouns are missing. Much more proofreading of the text is recommended as part of the overall revisions.

We have gone through the whole manuscript to improve the quality of the text. We hope it is now easier to read.

b) Bring forth the significance/importance/need for their work earlier. For example, the last section of the Discussion ('importance of this work') could be moved to the beginning of the paper.

We have kept the last section where it was but we added some sentences emphasizing importance of this work in the Introduction.

c) Edit and expand writing to be more accessible, to be more understandable to the general reader, and to avoid confusion in what is meant.Specifically: i) Since this paper introduces a new methodology, and most readers will not have a e.g. machine learning background, I believe the paper needs to improve the explanation of the methodology, and not just in the Materials and methods, but throughout the paper. Clearly the authors have already attempted to make it readable to non-experts, but I would specifically recommend the following:- At the beginning of the results, a summary of the method should be included. That is, present the method in a "self-contained" way so that the reader does not need to 'walk through' several methods (kCSD, sCSD, skCSD) and remarks on the innovations in each "iteration" as given in the Materials and methods at end of the paper.- It should be possible to read the paper from beginning to end without running into terms that have not been defined yet (e.g. subsection “Dependence of reconstruction on noise level”: 'regularization'; subsection “Dependence of reconstruction on the number and arrangement of Recording Electrodes”, third paragraph: 'the width of the basis functions'; 'the cross-validation error').- In the Materials and methods: some intuition for the kernel method should be given (and not by referring to another paper). Why would one 'avoid direct estimation of coefficients a_j'?. What is gained with the kernel trick? One will have to estimate fewer parameters, but how does that come about? Does this rely on specific assumptions?

We added a new section “Single cell kernel Current Source Density method, a high-level overview” at the beginning of Results where we address these points. We feel rather satisfied with the way we have addressed the first two points above, less so with the third. We found it challenging to build simple intuitions of why the kernel methods work as they do. However, we are willing to further expand this point if the reviewers feel this would benefit the readers.

ii) Is it meaningful to make comparisons to interpolated voltage method or kCSD method? These methods are not, to my knowledge, designed or expected to have resolution at the single cell (or subcellular level) – well, I suppose the resolution is determined by the electrode grid, but in any case it is not clear to me that readers would assume that these methods "should" be able to resolve branch points or other information local to a dendritic tree. Edit/expand the manuscript to make clear what is meant.

We added a paragraph discussing this point. We feel that without the method proposed here, the most natural approach to analyze current sources is through use of the regular, population CSD. Such an approach was used, for example, to investigate the changing distribution of current sources during action potential generation using data from high-definition MEA (Frey et al., 2009). What we show is that while CSD (kCSD) and skCSD are consistent, using the additional information about morphology renders significantly more detail of the activity studied.

iii) “Proof of concept experiment”. I am confused by this section. The authors go into detail describing the spatial and dynamical features of the inferred transmembrane currents. But then they seem to nullify their observations and the usefulness of their method when they write "From an experimental setup consisting of only 14 electrodes on a linear probe a detailed distribution of current sources along a complex morphology cannot be expected". Then, they write "but the firing activity is well observable" – but this point is not novel or contested. Indeed, spike identification from multi-electrode recordings is a common data analysis task, and not the (apparent) point of the method presented here.See also paragraph in Discussion beginning: "The skCSD method performed adequately for the proof of concept experimental data". I don't feel this is justified, without further clarification of the message in the subsection “Proof of Concept experiment: Spatial Current Source Distribution of Spike-triggered Averages”. Edit/expand the manuscript to make clear what is meant.

We have rewritten this section. Part of confusion stemmed from of our attempt to indicate the limitation of the method when applied to data from a small number of electrodes. We now added a new section in the Discussion, General observations, where we address this and other points raised by the referees.

iv) “To test the effect of branching on the results, a simple Y-shaped morphology was used (Figure 11). […] The first was stimulated at 5, 45, 60 ms, the other at 5, 25, 60 ms after the onset of the simulation.” Why choose these arbitrary stimulation times? Why are the synapses activated in this specific temporal sequence? It would be interesting to know what the temporal resolution of the technique is. Also, how would the performance of the technique change for different degrees of correlation of synaptic inputs? Edit/expand the manuscript to address reasoning/rationale of these questions.

The idea here was to consider the inputs stimulated together and separately. The times of activation were randomly selected in such a way to leave enough time for the membrane activity to settle down. This can be viewed as extreme cases of correlated and uncorrelated events. The skCSD reconstruction is not affected directly by the temporal correlation of the synaptic inputs. Just like any CSD estimation method, skCSD is applied to the potentials recorded at a given point in time. Of course, it is affected indirectly in the sense that slower or faster oscillating inputs lead to different spatial patterns due to filtering effects of the dendritic membrane: fast oscillations induce short dipoles, slow oscillations allow the current to spread along the cell leading to stronger dipoles. We address these effects indirectly with the figure studying the reconstruction of Fourier CSD patterns of increasing spatial frequency. We mention it now in the text.

v) “As a realistic example, we used a mouse retinal ganglion cell morphology Kong et al. (2005) from NeuroMorpho.Org Ascoli (2006). […] The cell was also driven with an oscillatory current. In the dendrites, only passive ion channels were used.” What is the frequency of the oscillatory input and its effect? They do say approximately 6.5Hz (line 175), but it would to helpful to know about different frequencies (say higher γ frequencies) and/or discuss it. How does the separation sensitivity of current sources depend on the frequency of the oscillatory input used? The validity of the technique should be tested for different oscillatory frequencies so that potential users keep in mind how the performance of current sources separation may vary as a function of input frequency. Is the separation of synaptic inputs equally successful for lower and higher synaptic input frequencies? As users will likely need to reveal current sources of different oscillatory input frequencies this information would be important to present. Edit/expand the manuscript to do and/or discuss these aspects.

This question relates to the previous one. While we feel this is an important question, we would rather not pursue this direction of study here, because we feel it would sidetrack the reader. It is true that different types of stimulation, be it by the experimenter or through correlated synaptic activity, will induce different spatio-temporal membrane activity. Since skCSD takes potentials at a given moment in time, what matters is a specific distribution of details that need to be resolved. From the perspective of the viability of the method it is secondary how exactly any specific distribution arose. We return to this point below illustrating it further with new figures.

vi) The Introduction (third paragraph) suggested to me that the current injection to the cell is exploited in this new method. However, this idea does not seem to be very central throughout the paper. Perhaps this could be stated more clearly.

This was a remnant of our initial discussions, an experiment we considered when starting the work on the method. We have now rewritten this part to better explain that what we really need is a morphology of a neuron and a set of simultaneous extracellular recordings, which may be obtained with different experiments.

3) Estimating parameters in real experiments.The authors suggest using a modeling approach to generate data and optimize the parameters and then use these parameters for the actual data.- First of all, from the Figure 8 results it seems the CV error is pretty good (though what is the CV error for case H?). Why not just use that? It would be informative if the authors illustrate how different the CSD estimates are when using the CV error instead of the L1 error in Figure 9.- My main concern with using the modeling approach (L1 error) is that the active properties of the neuron are not known sufficiently well. But I would expect that the presence of active currents (e.g. sodium, calcium, NMDA spikes, but also subthreshold active currents) would have a significant impact on the parameter estimates (R, λ etc.), compared to the parameters for a passive model (see suggestion in the subsection “Experimental Recommendations”). The authors should explore this, i.e. show how parameter estimates vary when considering a passive versus some variations of active models. There are various detailed hippocampal pyramidal cell models in modelDB from which the authors could use the active properties for their cell reconstruction and compare the obtained skCSD parameters with those obtained from a passive model.

Regarding the first point. Indeed, that is a good point. We did consider that, however, our studies indicated overall better results for the approach we recommended (L1). It is true that often the differences were not so great, so in experimental practice it might well be easier without much loss to apply CV error for parameter selection. Indeed, that is what we often do in kCSD applications. We have modified the text to address this. In Author response image 1we include figures from our original studies, left for L1, right for CV error, for the same moment of the cell model activity as in the paper; ignore the errors shown in the corner of each figure, they are not compatible (the rights shows L1 error – similarity – for the solutions obtained with CV). There is more. However, which can be done with considering different CSD distributions on a morphology, computing extracellular potential and reconstructions. This procedure allows one to better understand which features are recovered and which are mingled.

We now address here the questions and comments from the second point but relate also to point iv and v above, as we believe the underlying phenomena are similar, all address the problem of reconstructing varying spatio-temporal electrophysiological mechanisms.

First observe, that in order to investigate the quality of reconstruction using L1 error one does not need to assume any specific composition of the membrane, ion channels, etc. We only need to place specific set of sources given by some function we specify. From that we can easily compute extracellular potential, which is enough to do the reconstruction which can in turn be compared with the original. This can be done many times for different GT distributions which allows one to study the quality of reconstruction for a given setup, which features are resolved, how well, etc. This is essentially what we do below. This was mentioned in the paper but we now have tried to explain it better.

Regarding your specific suggestion, the advantage of using regular simulation, with specific composition of the membrane, has obvious advantages as it allows to simulate more physiological distributions. The disadvantage is that we typically do not know the model composition. Although there are plenty neuron models available for download with more complex membrane current mechanisms than we used, a proper study of the various phenomena you mentioned would have exceeded our capacity within the resubmission time. The main differences in the examples presented in the paper and those which might arise in simulations with e.g. various oscillatory input currents, ion-channels and NMDA spikes, would originate from the broader range of spatio-temporal variability in membrane current source density distribution.

We argued earlier, that the temporal resolution depends only on the sampling frequency of the electrode recording system, which is why we focus on the effects of the spatial resolution. To this end we distributed different Gaussian current source patterns on both branches and the stem of the Y-shaped neuron using different parameters: height, width and location, with one Gaussian source on each branch. The motivation was to simulate a wide range of membrane current patterns, which could represent diverse membrane processes. The temporal evolution of these patterns was not taken into account as the spatial resolution is independent. The reconstructions and the ground truth are shown in the interval representation of the 40 patterns along the segments. Author response image 2 shows the ground truth membrane current distribution, reconstructions from 4x4 and 4x16 electrodes and the calculated L1 errors for each pattern. One set of parameters was selected based on the L1 error for all the patterns and although with more electrodes the skCSD method performs better, narrow patterns are nevertheless strongly smoothed in space (e.g. patterns 27-30). In conclusion, there is a limit to the spatial resolution of the method, membrane current mechanisms with peaky spatial distributions will appear as wider but lower amplitude phenomena.

**Author response image 2. respfig2:** 40 current source density distribution patterns on the Y-shaped morphology: ground truth, skCSD reconstruction from 4x4 and 4x16 electrodes, and the L1 error for each pattern. The optimal parameters for the skCSD reconstruction were chosen by cross-validation for the whole series of patterns. The reconstruction error is higher for patterns with sharp peaks, in this case the skCSD method smooths these peaks in space.

- In general, the effects of the parameters λ, R and M are not properly discussed. How strongly do they affect the CSD estimates?

In general, the larger the basis the better. Ideally we would like to have infinite basis with Gaussians distributed smoothly parametrized with position on the morphology. However, it is computationally expensive to use a large basis, so there is a trade-off between the number of CSD features we wish to reconstruct and our patience and computational power. The width of the basis function also has an impact on the spatial resolution of the method, so in the paper we focused on optimising only one of these parameters. To investigate reconstruction quality in the parameter space set by the number of basis functions (M), basis function width (R) and regularization parameter (λ), we used the simulation setup for Y-shaped morphology with 4x4 electrodes as described in the paper. Author response image 3 shows the color-coded L1 reconstruction error for M = 32, 128, 512, 1024 when R = 8, 16, 32, 64, 128, and λ = 1e-5, 1e-4, 1e-3, 1e-2, 1e-1. As expected, 32 basis functions are not enough for a good estimation, but in case of 512 and 1024 basis functions the error map gets similar, although the latter case provides slightly better estimation.

**Author response image 3. respfig3:** We used the Y-shaped morphology and the 4x4 electrode setup to investigate the effect of using various basis numbers for the reconstruction. L1 error was calculated to compare the results for basis with 32, 128, 256, 512, 1024 elements (M), for several values of basis width (R) and λ. As shown, using few and narrow basis functions can lead to poor reconstruction, but covering the morphology with a sufficient number of basis functions of reasonable width (on the order of L/M with L being the total dendritic length) significantly improves the reconstruction (reduces the error).